# RESPECT THE MODEL:
# FINE-GRAINED AND ROBUST EXPLANATION WITH SHARING RATIO DECOMPOSITION

**Sangyu Han**[*], **Yearim Kim**,[*] **Nojun Kwak**[†]
Seoul National University
Seoul, 08826, Korea
{acoexist96,yerim1656,nojunk}@snu.ac.kr

## ABSTRACT

The truthfulness of existing explanation methods in authentically elucidating the underlying model's decision-making process has been questioned. Existing methods have deviated from faithfully representing the model, thus susceptible to adversarial attacks. To address this, we propose a novel eXplainable AI (XAI) method called SRD (Sharing Ratio Decomposition), which sincerely reflects the model's inference process, resulting in significantly enhanced robustness in our explanations. Different from the conventional emphasis on the neuronal level, we adopt a vector perspective to consider the intricate nonlinear interactions between filters. We also introduce an interesting observation termed Activation-Pattern-Only Prediction (APOP), letting us emphasize the importance of inactive neurons and redefine relevance encapsulating all relevant information including both active and inactive neurons. Our method, SRD, allows for the recursive decomposition of a Pointwise Feature Vector (PFV), providing a high-resolution Effective Receptive Field (ERF) at any layer.

## 1 INTRODUCTION

In light of the remarkable advancements in deep learning, the necessity for transparent and reliable decision-making has sparked significant interest in explainable AI (XAI) methods. In response to this imperative demand, XAI researchers have aimed to provide insightful and meaningful explanations that shed light on the decision-making process of complex deep learning models. However, the reliability of existing explanation methods in providing genuine insights into the decision-making process of complex AI models has been questioned.

Previous methods have not consistently adhered to the model but rather customized it to their respective preference. As a result, many of them are vulnerable to adversarial attacks, causing doubt on their reliability. To address this issue, we focus on faithfully representing the model's inference process, relying exclusively on model-generated information, and refraining from any form of correction. This approach supports the robustness of our explanations compared to other methods.

Moreover, existing methods have traditionally analyzed models at the neuronal level, often overlooking the intricate nonlinear interaction between neurons[1] to form a concept. This approach has been derived from the assumption that an individual scaler-valued channel (a filter or a neuron) carries a specific conceptual meaning. That is, the value of a single neuron directly determines the conceptual magnitude with the significance of a pixel being determined as a linear combination of each constituting neuron's conceptual magnitude. However, this assumption may oversimplify the complex nature of deep learning models, wherein multiple neurons nonlinearly collaborate to form a concept. Therefore, we analyze the models from a perspective of a *vector*, exploring the vector space to account for the interaction among neurons. Specifically, we introduce the pointwise feature vector (PFV), which is a vector along a channel axis of a hidden layer, amalgamating neurons that share the same receptive field.

---

[*]Equal contribution. [†]Corresponding author.
[1]A neuron outputs a scalar, an element in a tensor, by combining the information in its receptive field.

In addition, we alter the conventional way of calculating relevance based on post-activation values into the one based on pre-activation values. It is widely believed that, for achieving conceptual harmony and class differentiation at the final layer, image activations from the same class should undergo progressive merging along the shallow to deep layers (Fel et al., 2023). With this belief, previous methods have primarily focused on analyzing the value of the post-activation output, identifying the key contributor to the merged concept. However, we observe a fascinating phenomenon termed Activation-Pattern-Only Prediction (APOP), which shows that classification accuracies can be considerably maintained without receiving any input image, relying solely on the on/off activation pattern of the network (Refer to Tab. 1 for details). This highlights the importance of not only considering active neurons but also inactive ones, as both contribute to forming the patterns. However, after the nonlinear activation process, such as ReLU, the information about the contributors to the inactive neurons is lost. Therefore, we consider the contribution of the neurons in the prior layer to inactive neurons to fully comprehend the contribution of features.

Table 1: Classification accuracies on ImageNet validation set achieving comparable performance without any inputs, solely relying on weights and the activation pattern. Here, activation pattern means that the model records masks where the inactive neurons are flagged, during a prediction. Even with an empty image, the model makes comparable predictions when ReLU and Maxpool are replaced by the recorded masks. More information about APOP is contained in Appendix D.

| | Top-1 | | Top-5 | |
|---|---|---|---|---|
| | *Original* | *APOP* | *Original* | *APOP* |
| VGG13 | .679 | .544 | .882 | .787 |
| VGG16 | .698 | .575 | .894 | .809 |
| VGG19 | .705 | .593 | .898 | .822 |
| ResNet18 | .670 | .487 | .876 | .734 |
| ResNet34 | .711 | .557 | .900 | .790 |
| ResNet50 | .744 | .569 | .918 | .794 |
| ResNet101 | .756 | .560 | .928 | .785 |
| ResNet152 | .769 | .612 | .935 | .826 |

Considering the aforementioned challenges, we present our novel method, Sharing Ratio Decomposition (SRD), which decomposes a PFV comprising preactivation neurons occupying the same spatial location of a layer into the shares of PFVs in its receptive field. Our approach is centered on faithfully adhering to the model, relying solely on model-generated information without any alterations, thus enhancing the robustness of our explanations. Furthermore, while conventional methods have predominantly examined models at the neuronal level, with linear assumptions about channel significance, we introduce a vector perspective, delving into the intricate nonlinear interactions between filters. Additionally, with our captivating observation of APOP, we redefine our relevance, focusing on contributions to the pre-activation feature map, where all pertinent information is encapsulated. Our approach goes beyond the limitations of traditional techniques in terms of both quality and robustness, by sincerely reflecting the inference process of the model.

By recursively decomposing a PFV into PFVs of any prior layer with our Sharing Ratio Decomposition (SRD), we could obtain a high-resolution Effective Receptive Field (ERF) at any layer, which further enables us to envision a comprehensive exploration spanning from local to global explanation. While the local explanation allows us to address *where* in terms of model behavior, the global explanation enables us to delve into *what* the model looks at. Furthermore, by decomposing the steps of our explanation, we could see a hint on *how* the model inferences (Appendix A).

## 2 RELATED WORKS

**Backpropagation-based methods** such as Saliency (Simonyan et al., 2014), Guided Backprop (Springenberg et al., 2015), GradInput (Ancona et al., 2018), InteGrad (Sundararajan et al., 2017), Smoothgrad (Smilkov et al., 2017), Fullgrad (Srinivas & Fleuret, 2019) generate attribution maps by analyzing a model's sensitivity to small changes through backpropagation. They calculate the error through backpropagation for the input value to indicate the importance of each pixel, often generating noisy maps due to the presence of noisy gradients. Furthermore, there is doubt on the credibility of these methods, claiming that the gradients are not used during the inference process.

In contrast, LRP (Bach et al., 2015) constructs saliency maps solely using the model's weights and activations, without gradient information. LRP calculates the contribution of every neuron by propagating relevance, while our method, SRD, calculates relevance of vectors. Yet, different from LRP families (Bach et al., 2015; Montavon et al., 2017; 2019), which either ignores or assigns minor contribution to negatively contributing neurons for the active neuron, we acknowledge the signifi-

cance of every contribution in the model's inference process. Moreover, while LRP may not account for contributions to inactive neurons, who hold vital information for the inference, we elaborately handle contributions to both active and inactive neurons.

**Activation-based methods** generate activation maps by using the linearly combined weights of activations from each convolutional layer of a model. Class Activation Mapping (CAM) (Zhou et al., 2016) and its extension, Grad-CAM (Selvaraju et al., 2017), enhance interpretability in neural networks by utilizing convolutional layers and global average pooling. Grad-CAM++ (Chattopadhay et al., 2018) further improves localization accuracy by incorporating second-order derivatives and applying ReLU for finer details. These CAM-based approaches assume that each channel possesses distinct significance, and the linear combination of channel importance and layer activation can explain the regions where the model looks at importantly. However, due to nonlinear correlations between neurons, the CAM methods, except LayerCAM, struggle at lower layers, yielding saliency maps only with low-resolution. In contrast, LayerCAM (Jiang et al., 2021) inspects the importance of individual neurons, aggregating them in a channel-wise manner. It seems similar to our SRD as it calcalates the importance of a pixel (thus a vector). However, it also disregards negative contributions of each neuron and does not account for contribution of inactive neurons, as gradients do not flow through them.

**Desiderata of explanations** The absence of a 'ground truth' poses challenges for objective comparisons, given that explainability inherently depends on human interpretation (Doshi-Velez & Kim, 2017). To mitigate this issue, specific desiderata have been established such as Localization, Complexity, Faithfulness, and Robustness (Binder et al., 2023). Localization demands accurate identification of crucial regions during model inference, while Complexity requires creating sparse and interpretable saliency maps. Faithfulness insists that the removal of 'important' pixels significantly impacts the model's prediction. Robustness necessitates consistent saliency maps under both random and targeted perturbations, ensuring resilience against manipulations aimed at misleading explanations (Ghorbani et al., 2019a; Dombrowski et al., 2019). Our model, SRD, surpasses other state-of-the-art methods in meeting these desiderata without any modification of neuronal contributions during model inference.

## 3 METHOD: SHARING RATIO DECOMPOSITION (SRD)

Our method provides the versatility to perform both in forward (Fig. 1) and backward (Fig. 2) passes through the neural network, enabling a comprehensive analysis from different angles. A formal proof demonstrating this equivalence is provided in Appendix C.

### 3.1 FORWARD PASS

**Pointwise Feature Vector** The pointwise feature vector (PFV), our new analytical unit, comprises neurons in the hidden layer that share the same receptive field along the channel axis. Consequently, the PFV serves as a fundamental unit of representation for the hidden layer, as it is inherently a pure function of its receptive field. For linear layers, we compute the contributions of the previous PFVs to the current layer directly, leveraging the distributive law. However, for nonlinear layers, it is challenging to obtain the exact vector transformed by the layer, leading us to calculate relevance instead. The output or activation of layer $l$, denoted as $A^l \in \mathbb{R}^{C \times HW}$, is composed of $HW$ PFVs, $v_p^l \in \mathbb{R}^C$, where $p \in \{1, \cdots, HW\} \triangleq [HW]$ denotes the location of the vector in the feature map. Note that each vector belongs to the same $C$-dimensional vector space $\mathcal{V}^l$.

**Effective Receptive Field** Each neuron can be considered as a function of its Receptive Field (RF), and likewise, other neurons situated at the same spatial location but within different channels are also functions of the same RF. Consequently, the PFV, which comprises neurons along the channel axis, serves as a collective representation of the RF, effectively encoding the characteristics of its corresponding RF. However, note that the contribution of individual pixels within the RF is not uniform. For example, pixels in the central area of the RF contribute more in the convolution operation compared to edge pixels. This influential region is known as the Effective Receptive Field (ERF), which corresponds to only a fraction of the theoretical receptive field (Luo et al., 2016). However, the method employed in (Luo et al., 2016) lacks consideration for the Activation-Pattern-Only Prediction (APOP) phenomenon and the instance-level of ERF. To address this limitation, we introduce

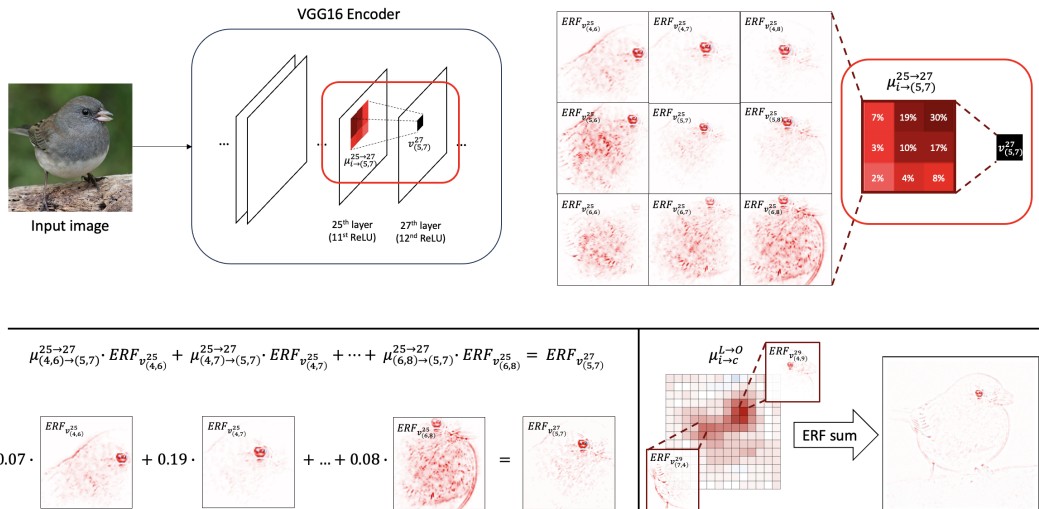

Figure 1: Forward Pass of our method. **Top**: An illustration of inference process. Red box portrays the contribution of $v_i^{25}$s in forming $v_{(5,7)}^{27}$, quantified by $\mu_{i \to (5,7)}^{25 \to 27}$. Each $v_i^{25}$ is labeled with its corresponding ERF. **Bottom Left**: The process of building ERF for $v_{(5,7)}^{27}$. **Bottom Right**: The final saliency map is derived as a weighted sum of the ERFs at the encoder output layer.

the sharing ratio $\mu$ to reflect the different contributions of pixels and make a more faithful ERF for each PFV. With our ERFs, we can investigate the vector space of the PFV, leading to a global explanation of the model. For more details, refer to Appendix A.

**Sharing Ratio Decomposition** Assuming we have prior knowledge of the sharing ratio, denoted as $\mu$, between layers (which can be derived at any point, even during inference), where $\mu$ signifies the extent to which each PFV contributes to the PFV of the subsequent layer (Exact way to obtain $\mu$ is deferred to Sec. 3.2). Given that we already possess information on the ERFs and sharing ratios of PFVs, we can construct the ERF of the next activation layer through a weighted sum of the constituent PFV's ERFs, expressed as follows (Fig. 1 Top, Bottom Left):

$$\sum_{l < k} \sum_i \mu_{i \to j}^{l \to k} \cdot ERF_{v_i^l} = ERF_{v_j^k}, \tag{1}$$

where $\mu_{i \to j}^{l \to k}$ is the sharing ratio of pixel $i$ of layer $l$ to pixel $j$ of the subsequent layer $k$, and $ERF_{v_i^l}$ is an ERF of PFV $v_i^l$. Note that we can summate the ERFs of different layers which are parallelly connected to the $k$-th layer, e.g., residual connection. For the first layer, its ERF is defined as:

$$ERF_{v_i^0} = E^i, \tag{2}$$

where $E^i$ is a unit matrix, where only the $i$-th element of the matrix is one, and all the others are zero. This means that the ERF for an input pixel is the pixel itself.

Consequently, we can sequentially construct the ERF for each layer until reaching the output of the encoder. The output consists of $HW$ PFVs along with their ERFs. The final saliency map $\phi_c(x)$ is obtained through a weighted sum of the ERFs of the encoder's output PFVs (Fig. 1 Bottom Right):

$$\sum_i \mu_{i \to c}^{L \to O} \cdot ERF_{v_i^L} = \phi_c(x), \tag{3}$$

where $\mu_{i \to c}^{L \to O}$ is the sharing ratio of the pixel $i$ of last layer $L$ to the pixel $c$ of the output (logit) $O$, which is the contribution of each PFV to output class $c$. As MLP classifier after encoder flattens the vectors to the scalars, there is no need to persist with our vector-based approach. Thus, for an MLP layer, we opt for the established backward methods such as Grad-CAM (Selvaraju et al., 2017).

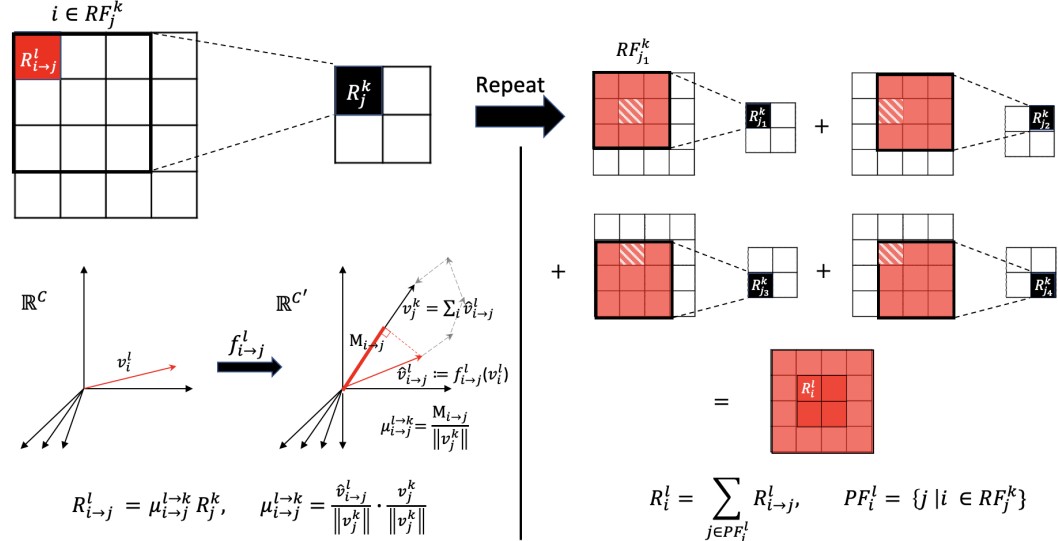

Figure 2: Backward Pass of our method. $i$ and $j$ are pixels in activation layer $l$ and $k$, respectively. **Left**: $v_j^k$ is a pre-activation PFV at activation layer $k$, $v_i^l$ is a post-activation PFV at activation layer $l$, $f_{i \to j}^l$ is an affine transformation function assigned to $(i, j)$. Summation of every $\hat{v}_{i \to j}^l$ leads to $v_j^k$ ($\sum_{i \in RF_j^k} \hat{v}_{i \to j}^l = v_j^k$). $\mu_{i \to j}^{l \to k}$ is a sharing ratio of each $v_{i \to j}^l$ to $v_j^k$. $R_{i \to j}^l$ is the relevance share of $i$ in the leading layer to $j$ in the following layer. **Right**: $RF_j^k$ is the receptive field of pixel $j$ and $R_j^k$ is the relevance score of $j$ to the output. Relevance $R_i^l$ in the leading layer can be calculated recursively using the next layer's relevance $R_j^k$'s via $R_{i \to j}^l$'s for $j$'s whose receptive field includes pixel $i$.

Additionally, in order to ensure class-discriminative saliency, we subtract the mean of its saliency and disregard any negative contributions. Then, the modified sharing ratio[2], $\mu_{i \to c}^{L \to O}$, for the encoder output layer is calculated as follows:

$$\mu_{i \to c}^{L \to O} = \max(\Phi_i^c - \frac{1}{K} \sum_{k \in [K]} \Phi_i^k, 0), \quad \Phi_i^c = \sum_k \alpha_k^c A_i^k, \quad \alpha_k^c = \frac{1}{HW} \sum_{i \in [HW]} \frac{\partial y^c}{\partial A_i^k}, \quad (4)$$

where $A_i^k$ is the $k$-th element of the PFV $v_i^{last}$ and $y^c$ is the $c$-th element of the output logit $y \in \mathbb{R}^K$ for $K$ classes. ReLU operation is ommited when calculating $\Phi_i^c$ since it is already applied after subtracting the mean.

## 3.2 BACKWARD PASS (FOR CALCULATING SHARING RATIO)

Suppose a PFV $v_j^k$ positioned at $j$ just prior to activation layer $k$. In a feed-forward network, $v_j^k$ is entirely determined by the $l$-th activation layer's PFVs $v_i^l$'s within the receptive field of $j$, $RF_j^k$, i.e,

$$v_j^k = f(V_j^{kl}) = \sum_i f_{i \to j}^l(v_i^l) = \sum_i \hat{v}_{i \to j}^l \quad \text{where} \quad V_j^{kl} = \{v_i^l | i \in RF_j^k\}, \quad (5)$$

for some affine function $f(\cdot)$ (See details in Appendix B). Note that PFV $v_j^k$ can be decomposed into $\hat{v}_{i \to j}^l$ which is a sole function of PFV $v_i^l$. In our approach, we initially define the relevance $R_j^k$ of pixel $j$ in layer $k$ as the contribution of the pixel to the output, typically the logit. Then, we distribute the relevance, $R_j^k$, to pixel $i$'s in layer $l$ by the sharing ratio $\mu_{i \to j}^{l \to k}$, which is calculated as taking the inner product of $\hat{v}_{i \to j}^l$ with $v_j^k$ and normalizing both vectors by $\|v_j^k\|$ as follows (Fig. 2 Left):

$$\mu_{i \to j}^{l \to k} = \langle \frac{\hat{v}_{i \to j}^l}{\|v_j^k\|}, \frac{v_j^k}{\|v_j^k\|} \rangle \text{ where } \hat{v}_{i \to j}^l = f_{i \to j}^l(v_i^l), \quad \text{i.e,} \quad \sum_{i \in RF_j^k} \mu_{i \to j}^{l \to k} = 1. \quad (6)$$

---

[2]The summation of modified sharing ratios does not necessarily equal to 1.

Then, according to the sharing ratio $\mu_{i \to j}^{l \to k}$, we decompose the relevance to the output:

$$R_{i \to j}^l = \mu_{i \to j}^{l \to k} R_j^k, \quad \text{i.e,} \quad R_j^k = \sum_{i \in RF_j^k} R_{i \to j}^l. \tag{7}$$

Finally, the relevance of $i$ to the output can be calculated as

$$R_i^l = \sum_{j \in PF_i^l} R_{i \to j}^l, \quad PF_i^l = \{j | i \in RF_j^k\}, \tag{8}$$

where $PF_i^l$ is the Projective Field of pixel $i$ to the next nonlinear layer (Fig. 2 Right).

The initial relevance at the last layer $L$, $R_{i \to c}^L$, is given as

$$R_{i \to c}^L = \mu_{i \to c}^{L \to O}, \tag{9}$$

where $\mu_{i \to c}^{L \to O}$ is the modified sharing ratio described in Eq. 4, which represents the contribution of pixel $i$ in the encoder output layer to class $c$.

## 4 EXPERIMENT

In this section, we conducted a comprehensive comparative analysis involving our proposed method, SRD, and several state-of-the-art methods: Saliency (Simonyan et al., 2014), Guided Backprop (Springenberg et al., 2015), GradInput (Ancona et al., 2018), InteGrad (Sundararajan et al., 2017), LRP$_{z+}$ (Montavon et al., 2017), Smoothgrad (Smilkov et al., 2017), Fullgrad (Srinivas & Fleuret, 2019), GradCAM (Selvaraju et al., 2017), GradCAM++ (Chattopadhay et al., 2018), ScoreCAM (Wang et al., 2020), AblationCAM (Ramaswamy et al., 2020), XGradCAM (Fu et al., 2020), and LayerCAM (Jiang et al., 2021).

In our experiments, we leveraged ResNet50 (He et al., 2016) and VGG16 (Simonyan & Zisserman, 2015) models[3] Each method has different choice of targeted layer for its best performance. Thus, we conducted experiments by targeting various layers to accomodate the varying resolutions of generated attribution maps. Since most CAM-based methods except for LayerCAM exhibit optimal performance when targeting higher layers, we generated low-resolution explanation maps for them. The dimensions of the resulting saliency maps were as follows: (7, 7) for low-resolution, (28, 28) for high-resolution, and (224, 224) for input-scale. All saliency maps were normalized by dividing them by their maximum values, followed by bilinear interpolation to achieve a resolution of (224, 224).

### 4.1 QUALITATIVE RESULTS

We visualize the counterfactual explanations of an original image with a cat and a dog. Fig. 3 shows that our explanations with SRD, are not only fine-grained but also counterfactual, while other methods do not capture the class-relevant areas and result in nearly identical maps. For more examples, refer to Appendix H.1.

### 4.2 QUANTITATIVE RESULTS

**Experimental setting** We conducted a series of experiments to assess the performance of our method compared to existing explanation methods. All evaluations were carried out on the ImageNet-S50 dataset (Gao et al., 2022), which contains 752 samples along with object segmentation masks.

**Metric** The metrics used in our experiments are as follows: To evaluate localization, Pointing Game (↑) (Zhang et al., 2018) measures whether maximum attribution point is on target, while Attribution Localization (↑) (Kohlbrenner et al., 2020) measures the ratio between attributions within the segmentation mask and total attributions. To evaluate complexity, Sparseness (↑) (Chalasani et al.,

---

[3]In this paper, we restrict our discussion to Convolutional Neural Networks (CNNs); however, the extension to arbitrary network architecture is straightforward.

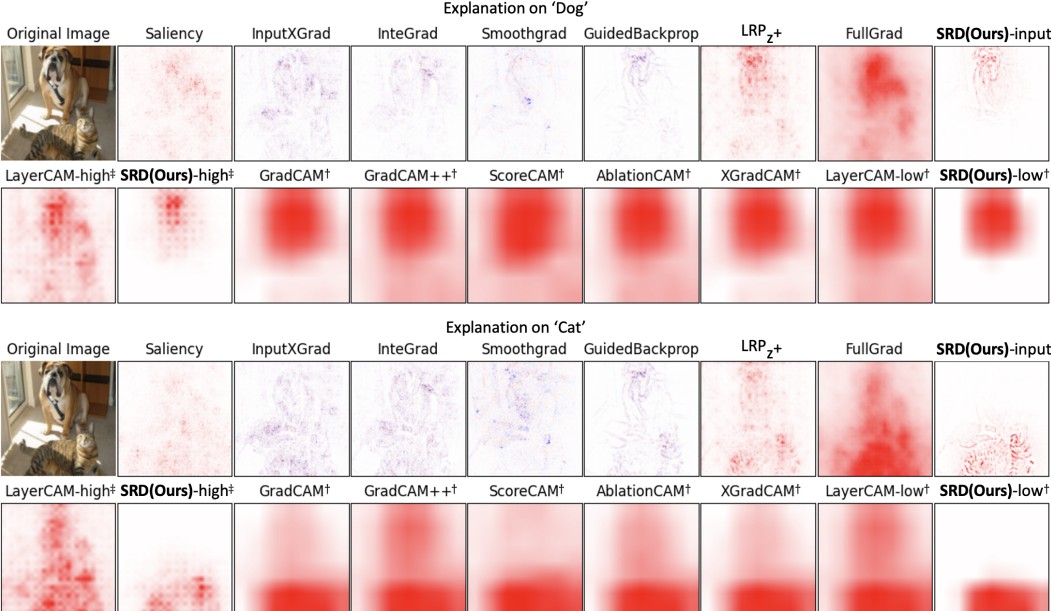

Figure 3: Qualitative results on ResNet50 for the class label 'Dog (**Top**)' and 'Cat (**Bottom**)'. Methods decorated with † have the resolution of (7, 7) and methods with ‡ have the resolution of (28, 28), while the others have the input-scale resolution, (224, 224). Notably, compared to other methods, SRD for input resolution is adept at capturing the fine details of the image. Best viewed when enlarged.

2020) measures how sparse the attribution map is, based on Gini index. For a faithfulness test, Fidelity (↑) (Bhatt et al., 2020) measures correlation between classification logit and attributions. To evaluate robustness, Stability (↓) (Alvarez Melis & Jaakkola, 2018) measures stability of explanation against noise perturbation, calculating the maximum distance between original attribution and perturbed attribution for finite samples. All of the metrics are calculated after clamping the attributions to [-1, 1], since all the attribution methods are visualized after clamping. Also, the arrow inside the parentheses indicates whether a higher value of that metric is considered desirable. For more details of the metrics, refer to Appendix E.

**Results** In the comprehensive evaluation presented in Table 2, our method, SRD, showcased superior performance across various metrics. Notably, for VGG16 architecture, SRD-high attained the highest scores in both the Pointing game and Fidelity, securing the second-highest score in Attribution Localization. Furthermore, SRD-input excelled in Sparseness and Stability, while consistently maintaining competitive scores across other metrics. This was particularly noteworthy when compared to input-scale methods.

In the case of ResNet, since many saliency map methods struggle to properly handle residual connections, some of the methods showed a decline in performance even when the model performance itself improved. Remarkably, our method retained its competitive performance on ResNet50. On ResNet50, SRD-high achieved the highest scores in Attribution Localization and Fidelity with the second highest score at Pointing game. Additionally, SRD-input achieved the best performance for Pointing game and Stability, achieving the second highest scores in Attribution Localization and Sparseness. These results point out that our proposed method, SRD, can give functional, faithful, and robust explanation.

## 4.3 ADVERSARIAL ROBUSTNESS

An explanation can be easily manipulated by adding small perturbations to the input, while maintaining the model prediction almost unchanged. This means that there is a discrepancy between the actual cues the model relies on and those identified as crucial by explanation. While the Stability metric in Sec 4.2 assesses the explanation method's resilience to random perturbations, Dombrowski

Table 2: Average results of Pointing game (Poi.), Attribution localization (Att.), Complexity (Com.), Sparseness (Spa.), Fidelity (Fid.), and Stability (Sta.) on Imagenet-S50 752 samples. All metrics are calculated after normalization, which is the default setting of Hedström et al. (2023). Methods decorated with † have the resolution of (7, 7) and methods with ‡ have the resolution of (28, 28), while the others have the input-scale resolution, (224, 224). We marked the highest result in **bold**, and the second with underline.

| Method | VGG16 | | | | | ResNet50 | | | | |
|---|---|---|---|---|---|---|---|---|---|---|
| | Poi.↑ | Att.↑ | Spa.↑ | Fid.↑ | Sta.↓ | Poi.↑ | Att.↑ | Spa.↑ | Fid.↑ | Sta.↓ |
| Saliency | .793 | .394 | .494 | .093 | .181 | .654 | .370 | .488 | .063 | .172 |
| GuidedBackprop | .892 | .480 | .711 | .022 | .100 | .871 | .498 | **.741** | .022 | .112 |
| GradInput | .781 | .387 | .630 | -.013 | .181 | .639 | .361 | .626 | -.018 | .178 |
| InteGrad | .869 | .416 | .618 | -.017 | .175 | .759 | .382 | .614 | -.016 | .171 |
| LRP$_{z+}$ | .855 | .456 | .535 | .098 | .182 | .543 | .332 | .572 | .012 | .105 |
| Smoothgrad | .845 | .363 | .536 | -.005 | .190 | .888 | .396 | .556 | -.002 | .166 |
| Fullgrad | .796 | .362 | .334 | .107 | .203 | .938 | .387 | .262 | .123 | .689 |
| GradCAM† | .945 | .431 | .466 | .175 | .583 | .946 | .424 | .411 | .128 | .757 |
| GradCAM++† | .932 | .429 | .351 | .176 | .570 | .945 | .414 | .386 | .129 | .732 |
| ScoreCAM† | .937 | **.582** | .342 | .167 | .622 | .916 | .381 | .313 | .123 | .827 |
| AblationCAM† | .928 | .481 | .493 | .189 | .622 | .934 | .394 | .329 | .133 | .814 |
| XGradCAM† | .896 | .406 | .446 | .181 | .576 | .946 | .424 | .411 | .126 | .753 |
| LayerCAM-low† | .869 | .425 | .446 | .175 | .450 | .934 | .411 | .379 | .128 | .734 |
| LayerCAM-high‡ | .865 | .435 | .401 | .199 | .423 | .941 | .423 | .349 | .135 | .486 |
| **SRD-low** (*ours*)† | .945 | .424 | .437 | .179 | .595 | .946 | .544 | .682 | .130 | .600 |
| **SRD-high** (*ours*)‡ | **.948** | .566 | .629 | **.206** | .406 | .952 | **.579** | .628 | **.142** | .375 |
| **SRD-input** (ours) | .925 | .561 | **.788** | .069 | **.099** | .953 | .576 | .724 | .082 | **.104** |

et al. (2019) evaluates the method's vulnerability to targeted adversarial attacks, while maintaining the logit output unchanged. The perturbation $\delta$ is optimized to minimize the loss below:

$$\mathcal{L} = \lambda_1 \left\| \phi(x_{adv}) - \phi(x_{target}) \right\|^2 + \lambda_2 \left\| F(x_{adv}) - F(x_{org}) \right\|^2, \tag{10}$$

where $x_{adv} = x_{org} + \delta$, $\phi(x)$ is the saliency map of image $x$, and $F(x)$ is the logit output of model $F$ given image $x$. We set $\lambda_1 = 1e11$ and $\lambda_2 = 1e6$ as in Dombrowski et al. (2019).

**Experimental setting** We conducted targeted manipulation on a set of 100 randomly selected ImageNet image pairs for the VGG16 model. Given that adversarial attacks can be taken only to gradient-trackable explanation methods, we selected Gradient, GradInput, Guided Backpropagation, Integrated Gradients, LRP$_{z+}$ and our SRD for comparison. The learning rate was 0.001 for all methods. For more detail, refer to the work of Dombrowski et al. (2019). The attack was stopped once the Mean Squared Error (MSE) between $x$ and $x_{adv}$ reached 0.001, while ensuring that the change in RGB values was bounded within 8 in a scale of 0-255 to let $x_{adv}$ be visually undistinguishable with $x$. After computing saliency maps, the absolute values were taken, as in Dombrowski et al. (2019). Since we obtained our $\mu_{i \to c}^{L \to O}$ by leveraging other methods, we set all $\mu_{i \to c}^{L \to O}$ to a constant value of 1 to eliminate the potential influence of other methods.

**Metric** To quantitatively compare robustness of the explanation methods towards the adversarial attacks, we measured the similarity between the original explanation $\phi(x_{org})$ and the manipulated explanation $\phi(x_{adv})$ using metrics such as the Structural Similarity Index Measure (SSIM) and Pearson Corelation Coefficient (PCC). High values of SSIM and PCC denote that $\phi(x_{adv})$ maintained the original features of $\phi(x_{org})$, thereby demonstrating the robustness of the explanation method.

**Result** In both PCC and SSIM results (Figure 4), SRD consistently outperformed other input-scale resolution saliency maps, attending the highest scores. his, coupled with the findings from the Stability experiments detailed in Table 2, substantiates that our proposed method, SRD, demonstrates exceptional resilience against adversarial attacks. Importantly, it maintains its explanatory efficacy even in the presence of perturbations, emphasizing its robustness.

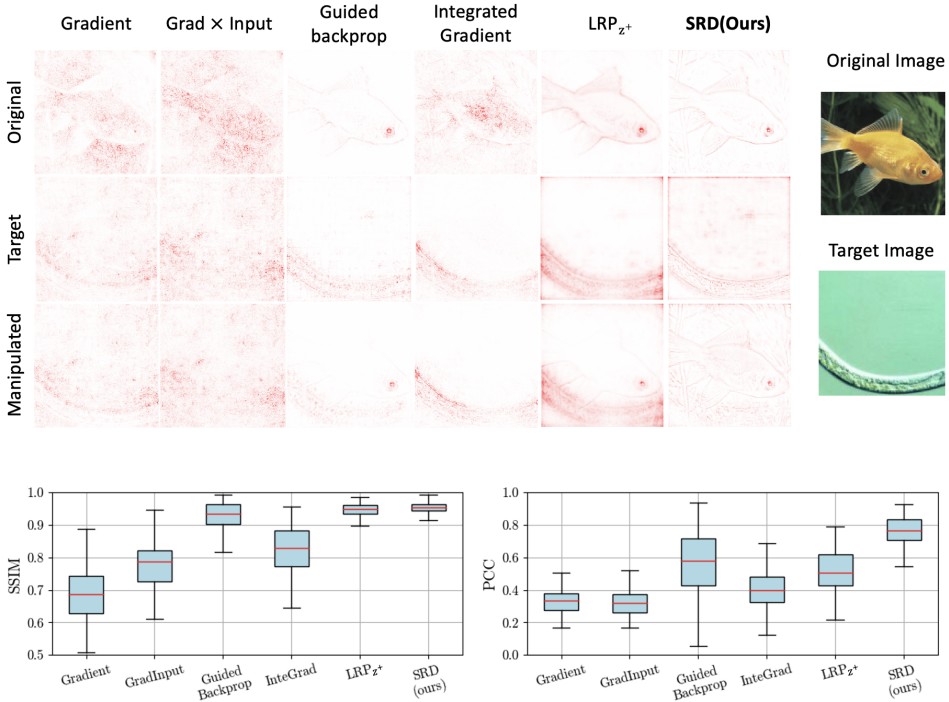

Figure 4: Adversarial attack experiment. Top: Qualitative comparison between explanations. While other methods deleted the goldfish (original image) in their explanation due to the manipulation, our method successfully retained the goldfish part. For more results, see Appendix H.2. Bottom: Quantitative result. Higher SSIM and PCC scores indicate less susceptibility to perturbation manipulation. In both SSIM and PCC, our method demonstrates superior defense against adversarial attack.

## 5 CONCLUSION

We propose a novel method, Sharing Ratio Decomposition (SRD), which analyzes the model with Pointwise Feature Vectors and decomposes relevance with sharing ratios. Unlike conventional approaches, SRD faithfully captures the model's inference process, generating explanations exclusively from model-generated data to meet the pressing need for robust and trustworthy explanations. Departing from traditional neuron-level analyses, SRD adopts a vector perspective, considering nonlinear interactions between filters. Additionally, our introduction of Activation-Pattern-Only Prediction (APOP) brings attention to the often-overlooked role of inactive neurons in shaping model behavior.

In our comparative and comprehensive analysis, SRD outperforms other saliency map methods across various metrics, showcasing enhanced effectiveness, sophistication, and resilience. Especially, it showcases notable proficiency in robustness, withstanding both random noise perturbation and targeted adversarial attacks. We believe that this robustness is a consequence of our thorough reflection of the model's behavior, signaling a promising direction for local explanation methods.

Moreover, through the recursive decomposition of Pointwise Feature Vectors (PFVs), we can derive high-resolution Effective Receptive Fields (ERFs) at any layer. With this, we would be able to generate a comprehensive exploration from local to global explanations in the future. Furthermore, we will go beyond answering *where* and *what* the model looks importantly to providing insights into *how* the model makes its decision.

## ACKNOWLEDGEMENT

This work was supported by NRF (2021R1A2C3006659) and IITP (2021-0-02068, 2021-0-01343) grants, all of which were funded by Korea Government (MSIT). It was also supported by AI Institute at Seoul National University (AIIS) in 2023.

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

## A   FUTURE WORKS: GLOBAL EXPLANATION WITH SRD

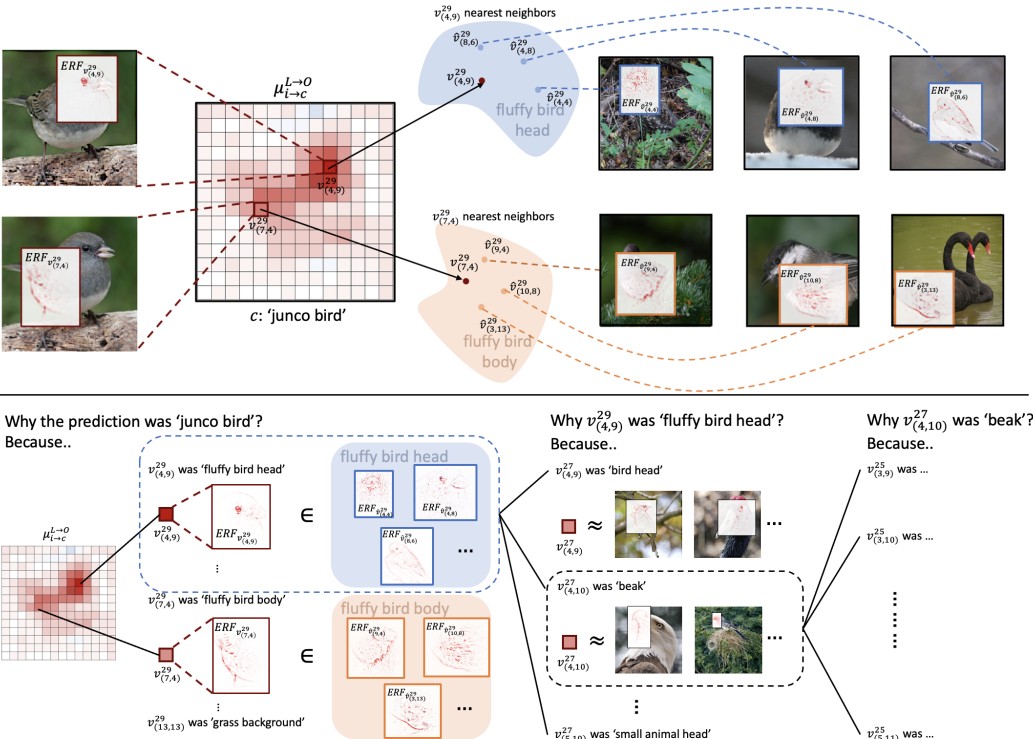

Figure 5: **Top**: Nearest neighbor PFVs encode similar concepts to that of the target PFV. By labeling each PFV with its ERF, we emphirically observed that the local manifold near certain PFV encodes a concept. For example, to know what $v_{(4,9)}^{29}$ encodes, we find its top 3 nearest neighbors from other samples' PFVs. **Bottom**: Recursive global explanation to explain the decision-making process of the model. Given modified sharing ratio, $\mu_{i \to c}^{L \to O}$, we know how much a certain concept of PFV $v_i^L$ at layer $L$ contributed to output. For example, $v_{(4,9)}^{29}$ is a PFV of (4, 9) in layer 29 which represents *"fluffy bird head"* concept contributed $\mu_{(4,9) \to c}^{L \to O}$ of the total prediction. $v_{(4,9)}^{29}$ is formed by the subconcepts of $[v_{(4,9)}^{27} : $ *"bird head"*$, v_{(4,10)}^{27} : $ *"beak"*$, ..., v_{(5,10)}^{27} : $ *"small animal head"*$]$, whose contributions are $\mu_{i \to (4,9)}^{27 \to 29}$. These 'subconcepts' can be further decomposed into minor concepts recursively, revealing the full decision-making process of the deep neural network.

Local Explanation methods explain *where* the model regards important for classification, and global explanation methods (Kim et al., 2018; Ghorbani et al., 2019b; Fel et al., 2023) explain *what* it means. However, with our method, SRD, we even go further to explain *how* the model makes its decision.

In short, we can provide *how the model predicted* along with *where the model saw* and *what it meant*. Through empirical observation, by labeling the Pointwise Feature Vector (PFV) with Effective receptive field (ERF), we discerned that each PFV encodes a specific concept. While there are numerous sophisticated global explanation methods available, for clarity, we opted for a more straightforward approach: examining the nearest neighbors of a given PFV. By observing its closest neighbors, we can discern the meaning of the target PFV (Top of Figure 5).

Furthermore, by analyzing the sharing ratio of a PFV, we gain insights into how each subconcept—components of the target PFV—shapes our target PFV (Bottom of Figure 5).

This recursive application allows SRD to thoroughly illuminate the model's decision-making process. Figure 5 shows an example of this attempt and gives a hint on how decision is made in a model. Detailed research on the global explanation of SRD will be dealt with in our next paper.

## B   DETAIL DESCRIPTION OF AFFINE FUNCTION $f_{i \to j}^{l}$

In this section, we describe how to calculate affine function $f_{i \to j}^{l}$ in Eq. 6.

**Convolutional layer** Each PFV in a CNN is transformed linearly by the convolutional layer and then aggregated with a bias term. We regard a PFV of a convolutional layer as a linear combination of PFVs in the previous layers in addition to the contribution of the bias vector. For example, consider a convolutional layer with a kernel $\omega \in \mathbb{R}^{C' \times C \times hw}$, where $C'$ and $C$ are the number of output and input channels, and $h$ and $w$ are the height and width of the kernel, respectively. The affine function of one convolutional layer $f_{i \to j}^{l}$ is defined as:

$$f_{i \to j}^{l}(v_i^l) = \omega_{i \to j} v_i^l + b \frac{\|v_i^l\|}{\sum_{k \in RF_j} \|v_k^l\|}, \tag{11}$$

where $\omega_{i \to j} \in \mathbb{R}^{C' \times C}$, the size of $RF_j$ is $h \times w$.

**Pooling layer** The average pooling layer computes the average of the PFVs within the receptive field. As a result, the contribution of each PFV is scaled down proportionally to the size of the receptive field. On the other hand, the max pooling layer performs a channel-wise selection process, whereby only a subset of channels from $v^l$ is carried forward to $v^{l+1}$. This is achieved by clamping the non-selected $v^l$'s contribution to zero:

$$f_{i \to j}^{l}(v_i^l) = \mathbb{1}_{m(v^l, j, i)} \odot v_i^l \tag{12}$$

Here, $\mathbb{1}_m \in \mathbb{R}^C$ is the indicator function that outputs 1 only for the maximum index among the receptive field, $RF_j$, which can be easily obtained during inference and $\odot$ denotes the elementwise multiplication. Thus, given information from inference, we can consider max pooling as a linear function, $f_{i \to j}^{l}$, whose coefficients are binary (0/1).

**Batch normalization layer** Additionally, for batch normalization layer, we manipulate each PFV in a direct manner by scaling it and adding a batch-norm bias vector to it, without resorting to any intermediate representation.

**Multiple functions** If there are multiple affine functions between $v_i^l$ and $v_j^{l+1}$, we composite multiple affine function along possible paths. For example, if there are max pooling layer and convolutional layers together, the resulting $f_{i \to j}^{l}$ would be:

$$f_{i \to j}^{l} = \sum_k g_{i \to k}^{l} \odot h_{k \to j}^{l}, \tag{13}$$

where $g_{i \to k}^{l}$ is affine function of max pooling layer and $h_{k \to j}^{l}$ is affine function of convolutional layer.

## C   PROOF OF EQUIVALENCE BETWEEN FORWARD AND BACKWARD PROCESSES

**Forward process:** Given that the saliency map for class $c$ being

$$\phi_c(x) = \sum_i \mu_{i \to c}^{L \to O} \cdot ERF_{v_i^L}, \tag{14}$$

and for each layer $l$ we have

$$ERF_{v_i^{l+1}} = \sum_j \mu_{j \to i}^{l \to l+1} \cdot ERF_{v_j^l}, \tag{15}$$

$ERF_{v_i^L}$ can be broken down as follows:

$$ERF_{v_i^L} = \sum_{j \in RF_i} \mu_{j \to i}^{(L-1) \to L} \cdot \Big( \sum_{k \in RF_j} \mu_{k \to j}^{(L-2) \to (L-1)} \big( \cdots \big( \sum_{p \in RF_q} \mu_{p \to q}^{0 \to 1} \cdot ERF_{v_p^0} \big) \big) \Big). \tag{16}$$

This can be generalized as

$$ERF_{v_i^L} = \sum_{p \in [HW]} \left(\sum_{\tau \in T} \prod_{l \in [L]} \mu_{p_{l-1} \to p_l}^{(l-1) \to l}\right) \cdot E^p, \tag{17}$$

where $\tau = (p_0 = p, p_1, \cdots, p_{L-1}, p_L = i)$ is a trajectory (path) from a pixel in an input image $p$ to a pixel in the last layer $L$, and $T$ denotes the set of all the trajectories. Note that $H$ and $W$ are the height and width of an input image and invalid trajectories have at least one zero sharing ratio on their path, i.e, $\mu = 0$ for some layer.

From Eq. 17, $\phi_c(x)$ becomes

$$\phi_c(x) = \sum_p \sum_i \mu_{i \to c}^{L \to O} \left(\sum_{\tau \in T} \prod_{l \in [L]} \mu_{p_{l-1} \to p_l}^{(l-1) \to l}\right) \cdot E^p. \tag{18}$$

**Backward process:** The saliency map $\phi_c(x)$ is defined as

$$\phi_c(x) = \sum_p R_p^0 \cdot E^p, \tag{19}$$

where

$$R_i^{l-1} = \sum_{j \in PF_i} \mu_{i \to j}^{(l-1) \to l} R_j^l \tag{20}$$

Thus, $R_0$ becomes

$$R_p^0 = \sum_{j \in PF_p} \mu_{p \to j}^{0 \to 1} \left(\sum_{k \in PF_j} \mu_{k \to j}^{1 \to 2} \left(\cdots \left(\sum_{i \in PF_q} \mu_{q \to i}^{(L-1) \to L} \cdot R_i^L\right)\right)\right). \tag{21}$$

This can be generalized as

$$R_p^0 = \sum_i R_i^L \left(\sum_{\tau \in T} \prod_{l \in [L]} \mu_{p_{l-1} \to p_l}^{(l-1) \to l}\right). \tag{22}$$

Since $R_i^L = \mu_{i \to c}^{L \to O}$ and $\phi_c(x) = \sum_p R_p^0 \cdot E^p$,

$$\phi_c(x) = \sum_p \sum_i \mu_{i \to c}^{L \to O} \left(\sum_{\tau \in T} \prod_{l \in [L]} \mu_{p_{l-1} \to p_l}^{(l-1) \to l}\right) \cdot E^p, \tag{23}$$

which is identical to Eq. 18.

## D   MORE RESULT OF APOP

We made an interesting observation during our experiments, which we term Activation-Pattern-Only Prediction (APOP). This phenomenon was discovered by conducting a series of experiments where a model made predictions with an image input. Subsequently, the model retained the binary on/off activation pattern along with its corresponding label (Algorithm 1). Following this, the model made a prediction once more, but this time with an entirely different input (*i.e. zeros, ones*) while keeping the activation pattern frozen.

All of our APOP experiments were conducted on the ImageNet validation dataset. We conducted experiments under three different input conditions: 'zeros', 'ones', and 'normal'. The 'zeros' setting is the experiment introduced in the main paper (Table 1). In 'ones' setting, we predicted again with matrix with ones instead of empty matrix. In 'normal' setting, matrix filled with normal distribution $N(0, 1)$ was used. As shown in Table 3, all settings achieved higher accuracy compared to random prediction baselines – 0.001 for Top-1 accuracy and 0.005 for Top-5 accuracy. Especially, it is intriguing that it achieved almost the same accuracy with the original accuracy in APOP & ReLU setting, supporting our idea that activation pattern is a crucial component in explanation, complementing the actual values of the neurons.

We carried out an additional experiment: Particular Layer Activation Binarization as illustrated in Figure 6. Instead of entirely freezing the activation pattern, we replaced the activation value of a

| Model | Input | Top-1 acc. | | | Top-5 acc. | | |
|---|---|---|---|---|---|---|---|
| | | Ori. | APOP | APOP & relu | Ori. | APOP | APOP & relu |
| VGG13 | zeros | 0.6790 | 0.5447 | 0.6603 | 0.8828 | 0.7870 | 0.8716 |
| | ones | | 0.4901 | 0.6594 | | 0.7390 | 0.8707 |
| | normal | | 0.2510 | 0.6580 | | 0.4513 | 0.8702 |
| VGG16 | zeros | 0.6980 | 0.5754 | 0.6847 | 0.8940 | 0.8094 | 0.8875 |
| | ones | | 0.5268 | 0.6837 | | 0.7665 | 0.8870 |
| | normal | | 0.2903 | 0.6854 | | 0.4979 | 0.8871 |
| VGG19 | zeros | 0.7052 | 0.5937 | 0.6948 | 0.8981 | 0.8226 | 0.8947 |
| | ones | | 0.5462 | 0.6937 | | 0.7801 | 0.8938 |
| | normal | | 0.2956 | 0.6957 | | 0.5067 | 0.8923 |
| Resnet18 | zeros | 0.6707 | 0.4871 | 0.6404 | 0.8769 | 0.7340 | 0.8595 |
| | ones | | 0.4813 | 0.6408 | | 0.6750 | 0.8593 |
| | normal | | 0.2518 | 0.6375 | | 0.4594 | 0.8598 |
| Resnet34 | zeros | 0.7113 | 0.5578 | 0.6917 | 0.9009 | 0.7906 | 0.8910 |
| | ones | | 0.5102 | 0.6902 | | 0.7456 | 0.8907 |
| | normal | | 0.3194 | 0.6918 | | 0.5380 | 0.8917 |
| Resnet50 | zeros | 0.7446 | 0.5690 | 0.7328 | 0.9183 | 0.7943 | 0.9148 |
| | ones | | 0.5198 | 0.7334 | | 0.7527 | 0.9141 |
| | normal | | 0.3035 | 0.7366 | | 0.5066 | 0.9168 |
| Resnet101 | zeros | 0.7560 | 0.5601 | 0.7459 | 0.9280 | 0.7853 | 0.9231 |
| | ones | | 0.5151 | 0.7431 | | 0.7433 | 0.9228 |
| | normal | | 0.3276 | 0.7514 | | 0.5379 | 0.9259 |
| Resnet152 | zeros | 0.7696 | 0.6124 | 0.7593 | 0.9359 | 0.8260 | 0.9304 |
| | ones | | 0.5585 | 0.7592 | | 0.7785 | 0.9300 |
| | normal | | 0.3561 | 0.7618 | | 0.5666 | 0.9319 |

Table 3: Additional results of APOP. Ori. is the original model, while APOP is the case where activation pattern of each activation layer is replaced (yet, the value of neurons can be negative). APOP & relu is the setting where after activation pattern is replaced, neurons are calculated with relu layer (every neuron has a non-negative value, while preserving information of the activation pattern).

particular layer into 1 or 0; if the activation value was greater than 0, then it was set to 1, otherwise, it was set to 0. Remarkably, even under this setting, the model predicted more accurately than random guessing. It happened even when this binarization occurred in the very first activation layer. This experiment reinforces our notion that the activation pattern holds comparable significance to the actual neuronal values.

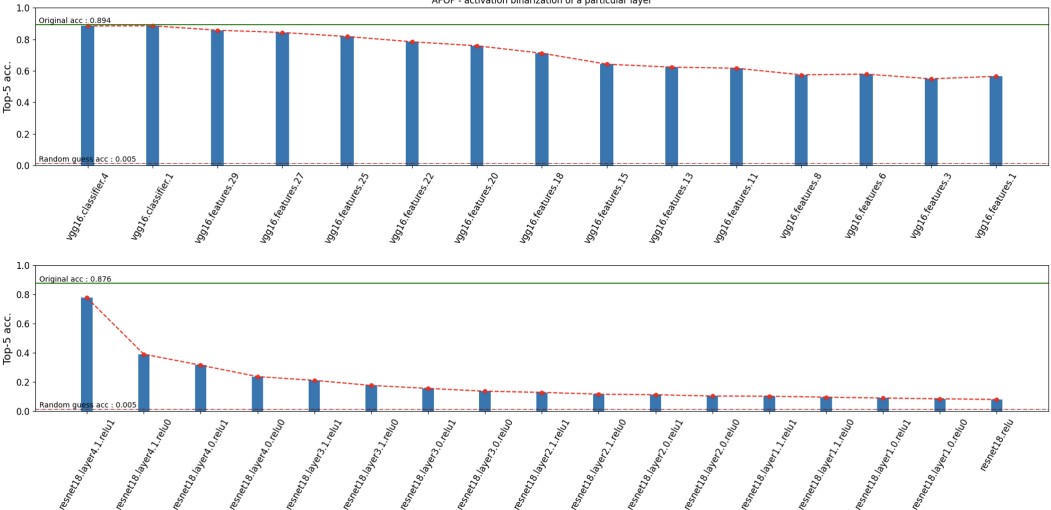

Figure 6: APOP - Particular Layer Activation Binarization. Target layer's activation was replaced with its binary version. In all layer, it achieved higher than random guessing baseline.

**Algorithm 1** APOP process in PyTorch pseudocode

```python
import torch
import torch.nn as nn
import torch.nn.functional as F

class CustomReLU(nn.ReLU):
    def forward(self,x):
        output = F.relu(x)
        self.mask = torch.sign(output) # make binary mask
        return output
    def APOP_forward(self,x):
        output = x * self.mask # mask inactive neuron
        return output

class CustomMaxPool2d(nn.MaxPool2d):
    def forward(self,x):
        output,self.mask_indices = F.max_pool2d(x,return_indices=True)
        return output
    def APOP_forward(self,x):
        output = indice_pool(x,self.mask_indices) # mask inactive neuron
                                                  # with saved mask_indices
        return output

total_sample = 0
original_correct_predictions = 0
APOP_correct_predictions = 0
model = CustomModel(model) # replace ReLU and Maxpool into CustomReLU and CustomMaxPool2d
empty_input = torch.zeros_like(data)
for data,labels in data_loader:
    original_predictions = CustomModel(x) # predict original prediction and save masks
    APOP_predictions = CustomModel.APOP_forward(empty_input) # APOP with saved masks
    original_correct_predictions += compute_accuracy(original_predictions,labels)
    APOP_correct_predictions += compute_accuracy(APOP_predictions,labels)
    total_samples += labels.size(0)

original_model_accuracy = original_correct_predictions / total_sample
APOP_model_accuracy = APOP_correct_predictions / total_sample
```

## E    DETAIL OF METRICS

**Pointing Game** (↑) (Zhang et al., 2018) evaluates the precision of attribution methods by assessing whether the highest attribution point is on the target. The groundtruth region is expanded for some margin of tolerance (15px) to insure fair comparison between low-resolution saliency map and high-resolution saliency map. Intuitively, the strongest attribution should be confined inside the target object, making a higher value for a more accurate explanation method.

$$\mu_{\text{PG}} = \frac{Hits}{Hits + Misses} \tag{24}$$

**Attribution Localization** (↑) (Kohlbrenner et al., 2020) measures the accuracy of an attribution method by calculating the ratio , $\mu_{\text{AL}}$, between attributions located within the segmentation mask and the total attributions. A high value indicates that the attribution method accurately explains the crucial features within the target object.

$$\mu_{\text{AL}} = \frac{R_{in}}{R_{tot}}, \tag{25}$$

where $\mu_{\text{AL}}$ is an inside-total relevance ratio without consideration of the object size. $R_{in}$ is the sum of positive relevance in the bounding box, $R_{tot}$ is the total sum of positive relevance in the image.

**Sparseness** (↑) (Chalasani et al., 2020) evaluates the density of the attribution map using the Gini index. A low value indicates that the attribution is less sparse, which may be observed in low-resolution or noisy attribution maps.

$$\mu_{\text{Spa}} = 1 - 2 \sum_{k=1}^{d} \frac{v_{(k)}}{||\mathbf{v}||_1} (\frac{d - k + 0.5}{d}), \tag{26}$$

where $\mathbf{v}$ is a flatten vector of the saliency map $\phi(x)$

**Fidelity** (↑) (Bhatt et al., 2020) measures the correlation between classification logit and attributions. Randomly selected 200 pixels are replaced to value of 0. The metric then measures the correlation

between the drop in target logit and the sum of attributions for the selected pixels.

$$\mu_{\text{Fid}} = \operatorname*{Corr}_{S \in \binom{[d]}{|S|}} \left( \sum_{i \in S} \phi(x)_i, F(x) - F\left(x_{[x_s = \bar{x}_s]}\right) \right),$$

where $F$ is the classifier, $\phi(x)$ the saliency map given $x$

**Stability** ($\downarrow$) (Alvarez Melis & Jaakkola, 2018) evaluates the stability of an explanation against noise perturbation. While measuring robustness against targeted perturbation (as discussed in Section 4.1) can be computationally intensive and complicated due to non-continuity of some attribution methods, a weaker robustness metric is introduced to assess stability against random small perturbations. This metric calculates the maximum distance between the original attribution and the perturbed attribution for finite samples. A low stability score is preferred, indicating a consistent explanation under perturbation.

$$\mu_{\text{Sta}} = \max_{x_j \in N_\epsilon(x_i)} \frac{\|\phi(x_i) - \phi(x_j)\|_2}{\|x_i - x_j\|_2}, \tag{27}$$

where $N_\epsilon(x_i)$ is a gaussian noise with standard deviation 0.1. all of the metrics are measure after clamping the attributions to [-1,1], as all the attrubution methods are visualized after clamping.

## F    ABLATION STUDY

To clarify the gains of our method, we conducted an ablation study for each factor (Figure 7). The scalar-based approach with our method can be regarded as LRP-0 (Bach et al., 2015). Next to it, we showcased the generated explanation when calculating the relevance with post-activation values. As you can see, compared to ours (SRD), the generated explanations with scalar are very noisy, while those with post-activation values are too sparse. With our observation of APOP, we have proven that we should consider every information including active and inactive neurons. This is the reason that we used vectors as our analysis unit and pre-activation values to propagate our relevance.

## G    APPLICATION TO VARIOUS ACTIVATIONS

| Activation | ReLU | ELU | LeakyReLU | Swish | GeLU | Tanh |
|---|---|---|---|---|---|---|
| GuidedBackprop | 0.064 | 0.025 | 0.001 | 0.015 | 0.030 | 0.028 |
| GradInput | -0.010 | -0.007 | -0.005 | -0.024 | -0.004 | -0.006 |
| InteGrad | 0.006 | 0.015 | -0.001 | -0.008 | -0.007 | 0.014 |
| LRP$_{z+}$ | 0.039 | - | - | - | - | - |
| Smoothgrad | -0.012 | 0.026 | -0.014 | -0.023 | -0.009 | -0.017 |
| Fullgrad | 0.038 | 0.209 | 0.029 | 0.171 | 0.095 | 0.107 |
| GradCAM | 0.005 | -0.014 | -0.004 | 0.042 | -0.001 | 0.002 |
| ScoreCAM | 0.013 | 0.052 | 0.031 | 0.061 | 0.010 | 0.017 |
| AblationCAM | 0.020 | 0.024 | 0.003 | 0.015 | 0.033 | 0.012 |
| XGradCAM | 0.007 | 0.011 | 0.018 | 0.028 | 0.012 | 0.017 |
| LayerCAM | 0.021 | 0.042 | 0.012 | 0.018 | 0.007 | -0.001 |
| SRD(Ours) | **0.078** | **0.214** | **0.065** | **0.194** | **0.128** | **0.115** |

Table 4: Fidelity results on various activation functions. We evaluated the fidelity metric of ResNet50 in CIFAR-100 with different activation functions: ReLU, ELU, LeakyReLU, Swish, GeLU, and Tanh. Our method, SRD, achieved highest performance on every activation function. The model accuracies with each activation varient were as follows: 0.780 for ReLU, 0.746 for ELU, 0.785 for LeakyReLU, 0.756 for Swish, 0.767 for GeLU, and 0.685 for Tanh.

Most of the existing methods have been limited to ReLU or have had to be redesigned for other activations. However, as in Fig. 8 and Tab. 4, SRD can be applied to various activations due to the utilization of preactivation, while maintaining high fidelity.

# H  ADDITIONAL SALIENCY MAP COMPARISON

## H.1  SALIENCY MAP COMPARISON

Fig. 9-18 are some examples that compare the saliency maps of different methods.

## H.2  EXPLANTION MANIPULATION COMPARISON

Fig. 19-23 are examples that compare explanation manipulation of different methods.

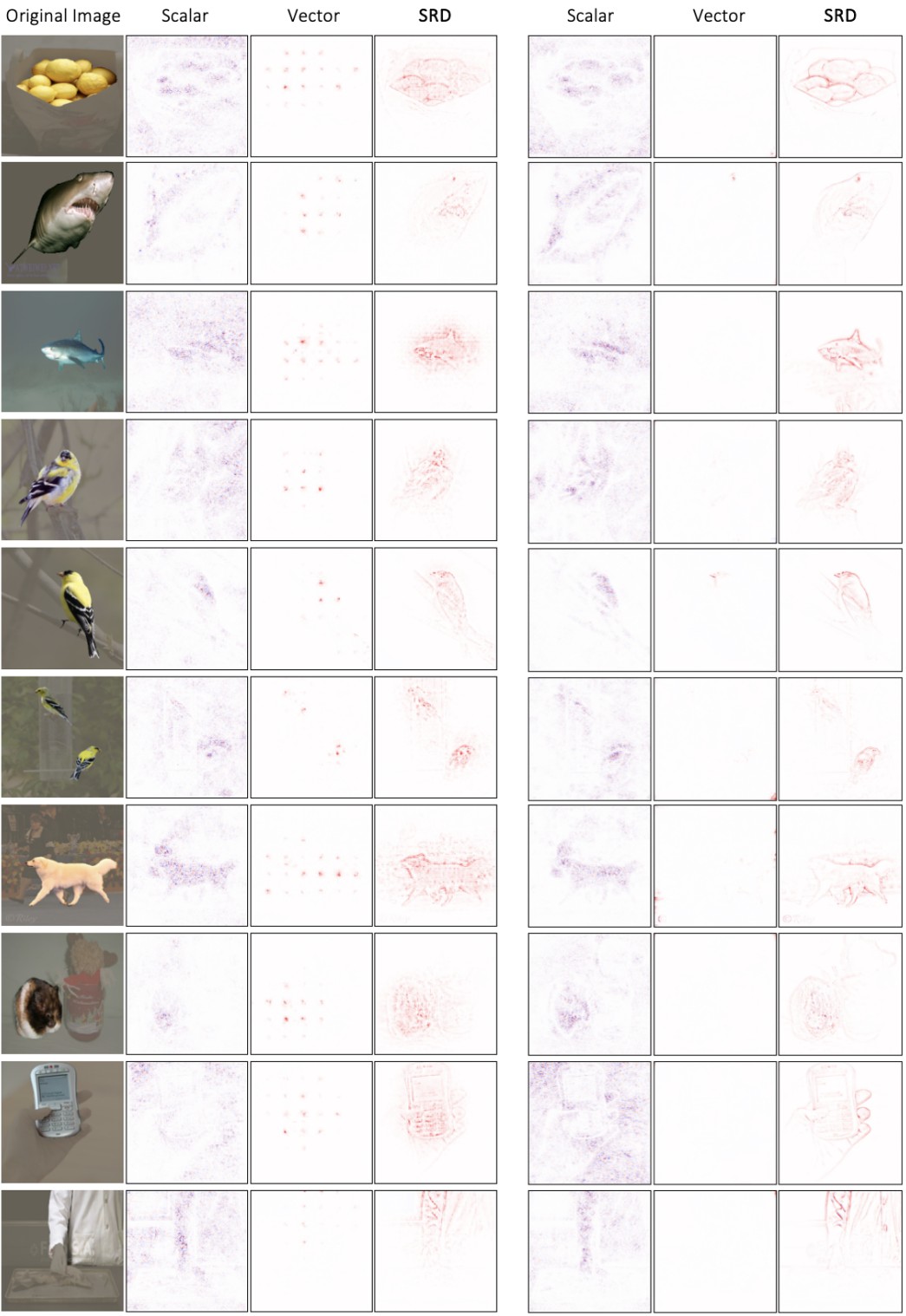

Figure 7: Ablation Study on ResNet50 (Left) and VGG16 (Right). We firstly generated the saliency maps with a neuron (scalar) rather than a vector as an analysis unit. And then, we analyzed with vectors, yet by using postactivation values. Lastly, we utilized vectors and pre-activation values, which is our method.

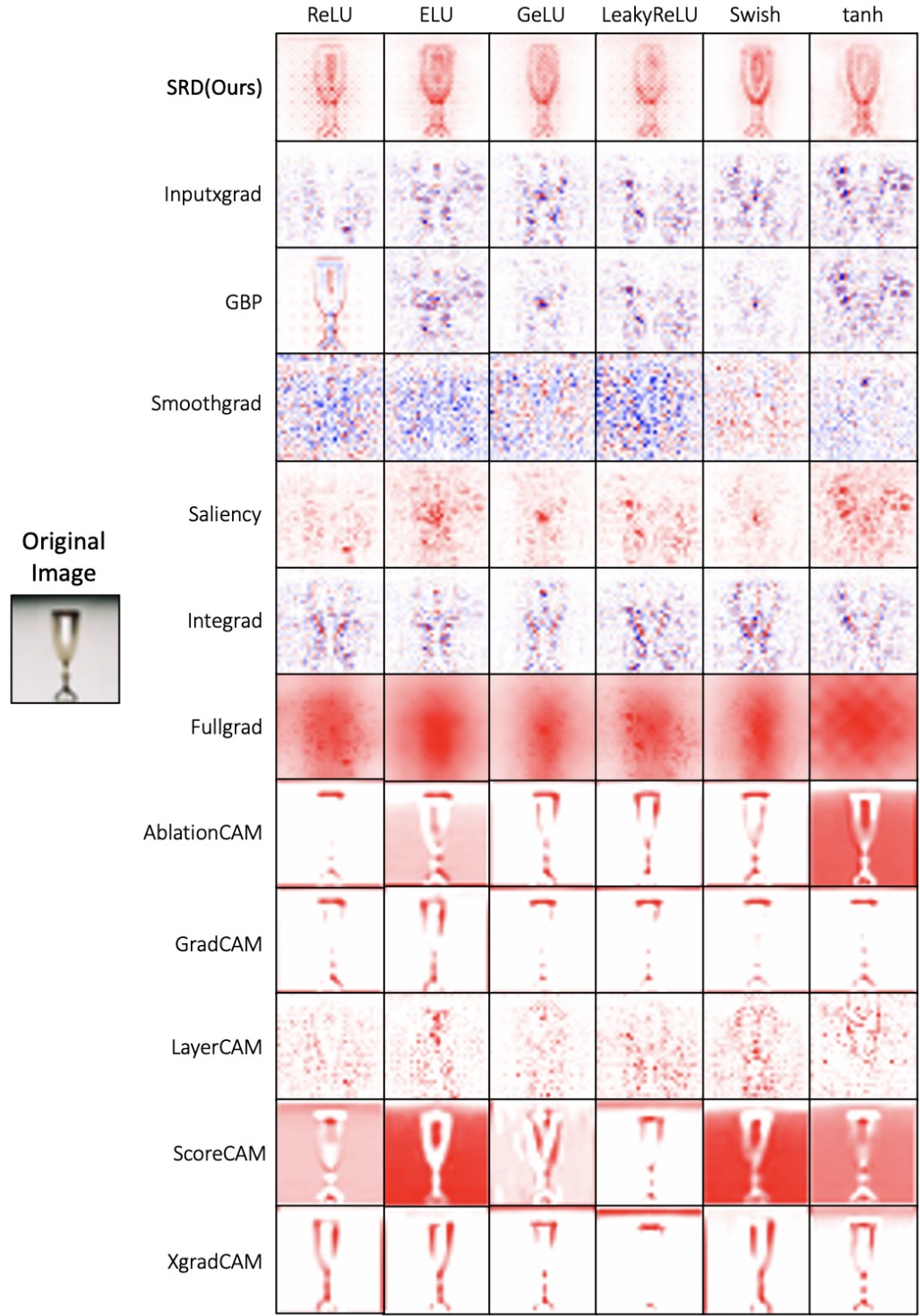

Figure 8: Qualitative results applied to various activation functions. Here, even with various activations, SRD generates the most fine-grained and feasible explanation maps.

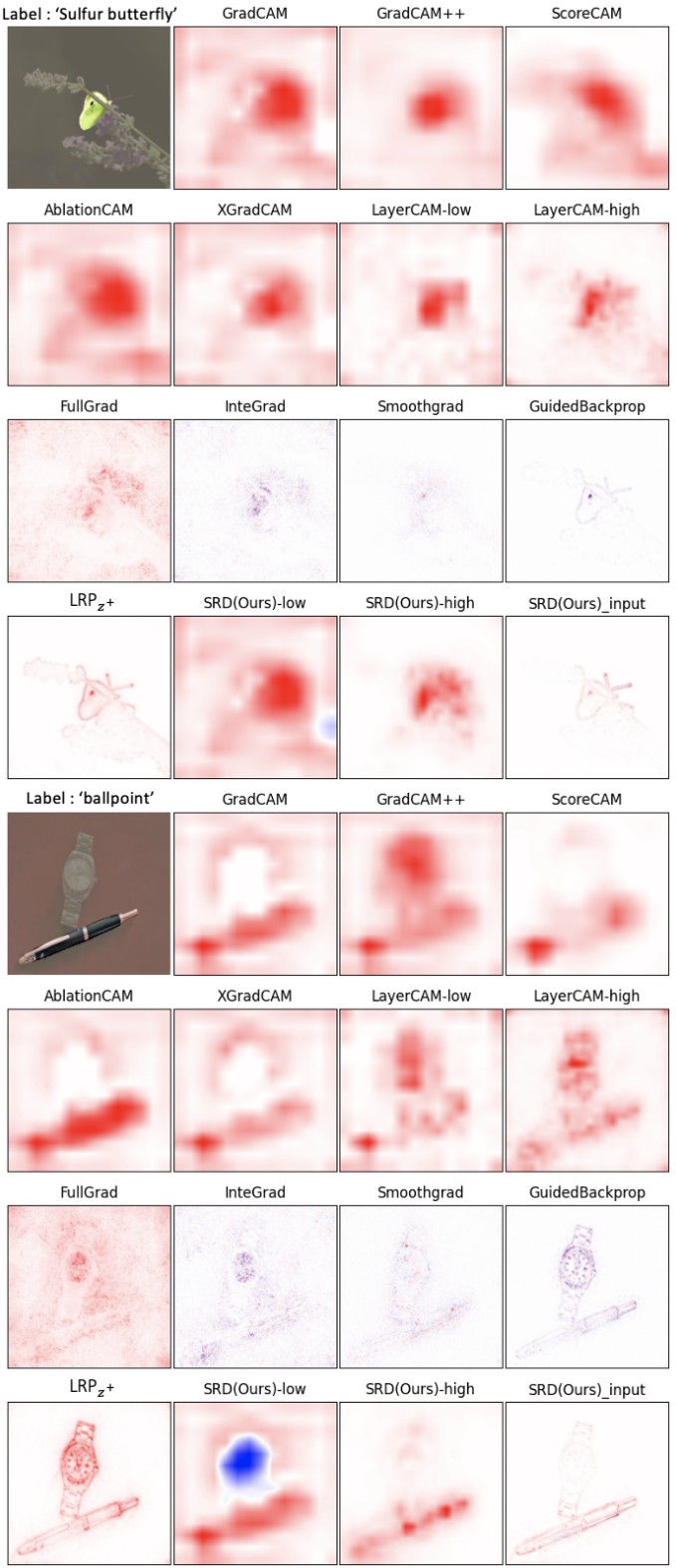

Figure 9: Qualitative comparison on VGG16. The highlighted region is the segmentation mask.

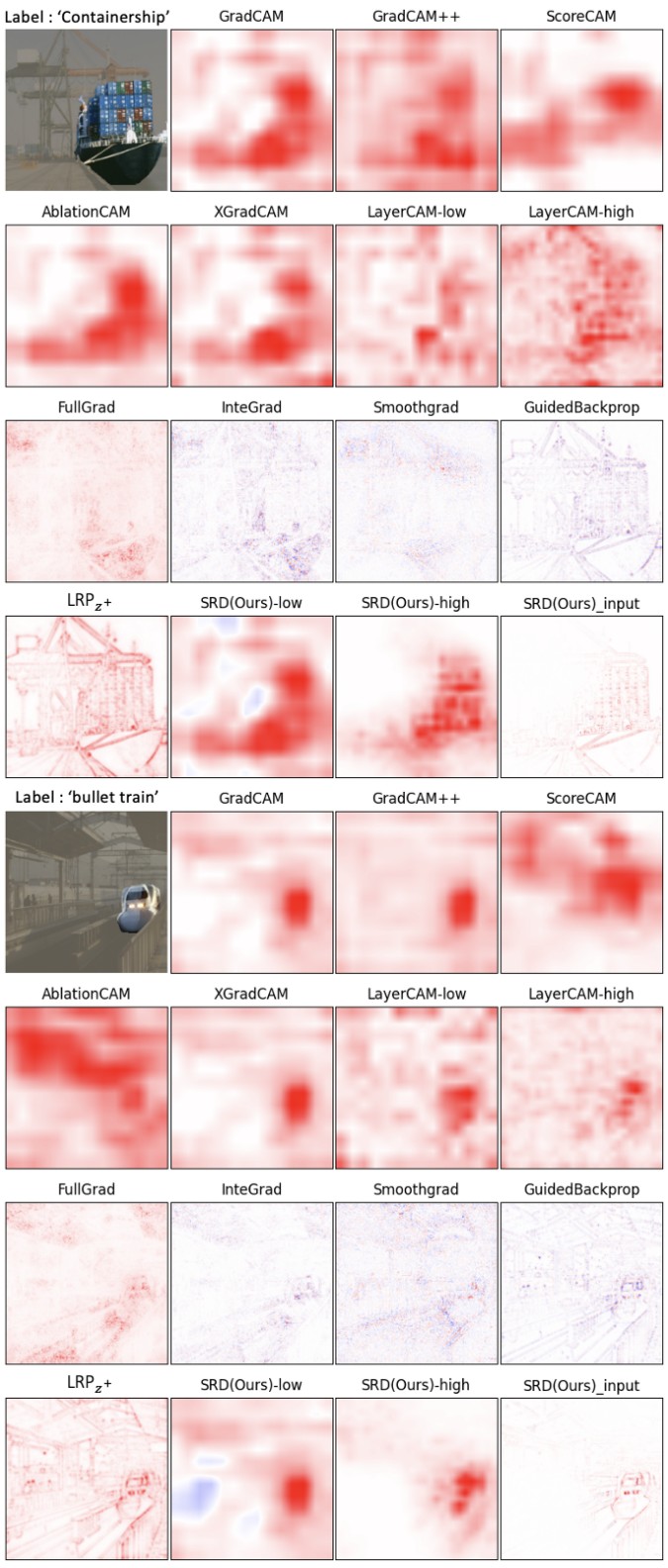

Figure 10: Qualitative comparison on VGG16. The highlighted region is the segmentation mask.

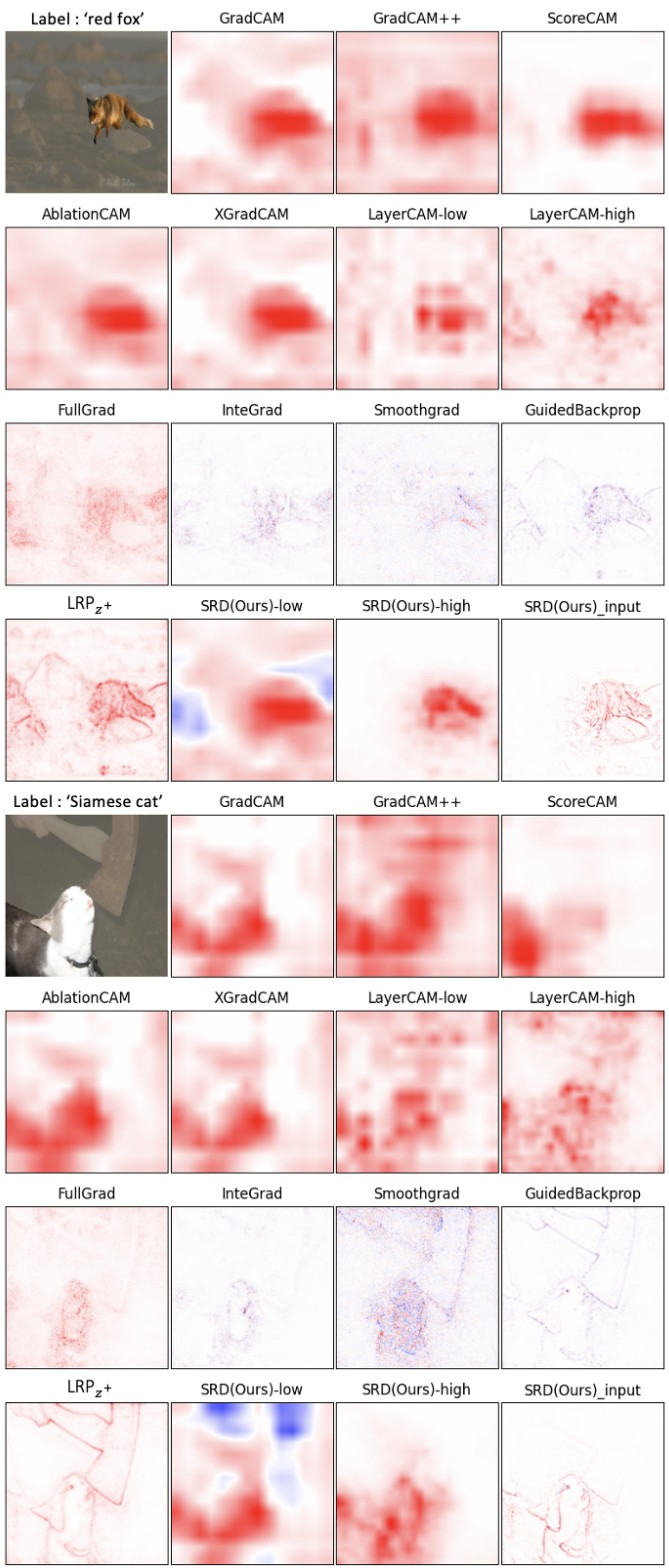

Figure 11: Qualitative comparison on VGG16. The highlighted region is the segmentation mask.

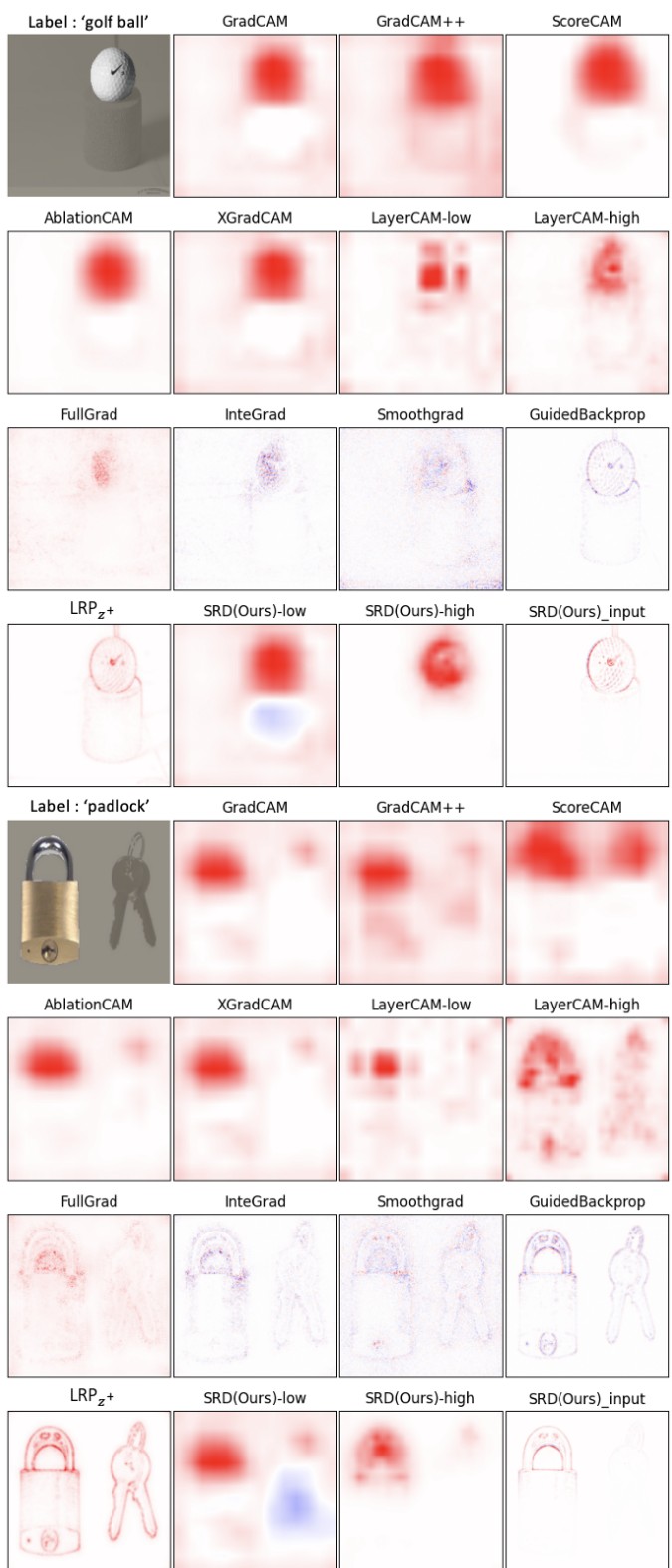

Figure 12: Qualitative comparison on VGG16. The highlighted region is the segmentation mask.

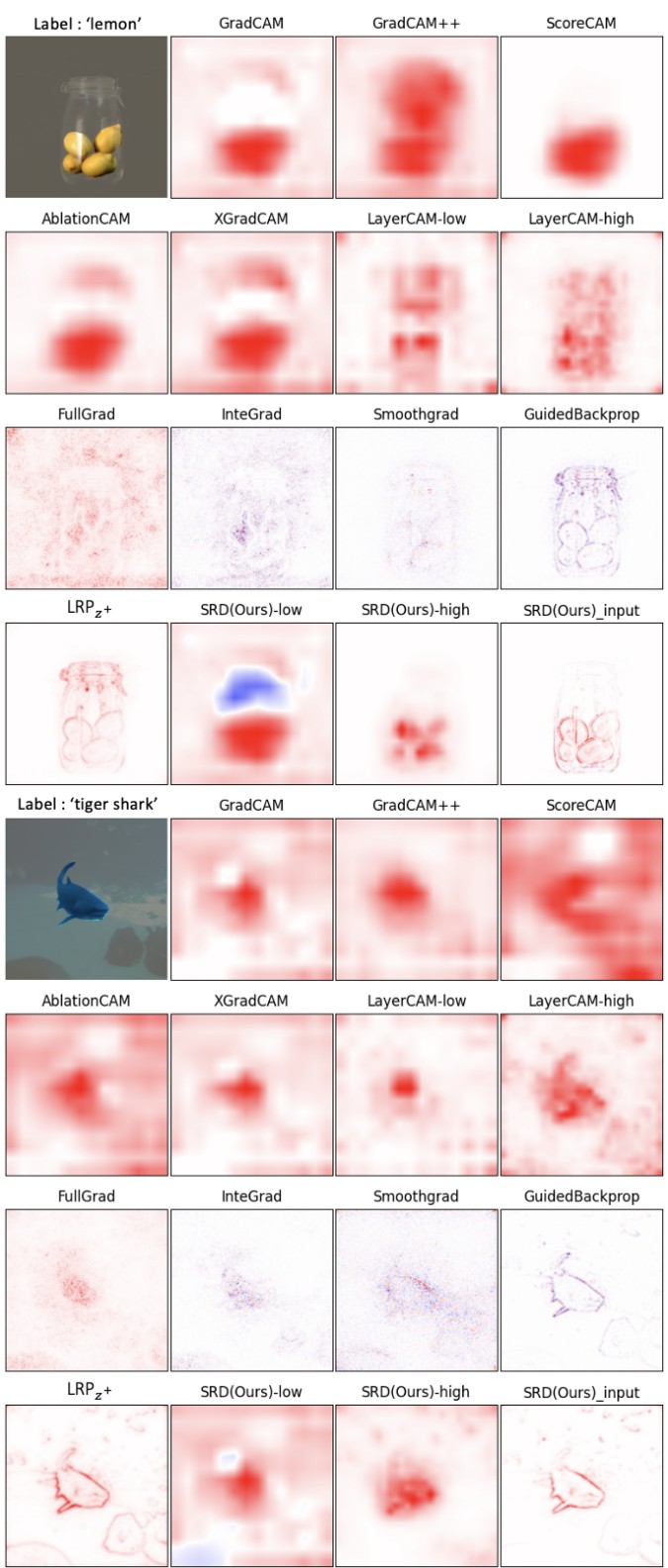

Figure 13: Qualitative comparison on VGG16. The highlighted region is the segmentation mask.

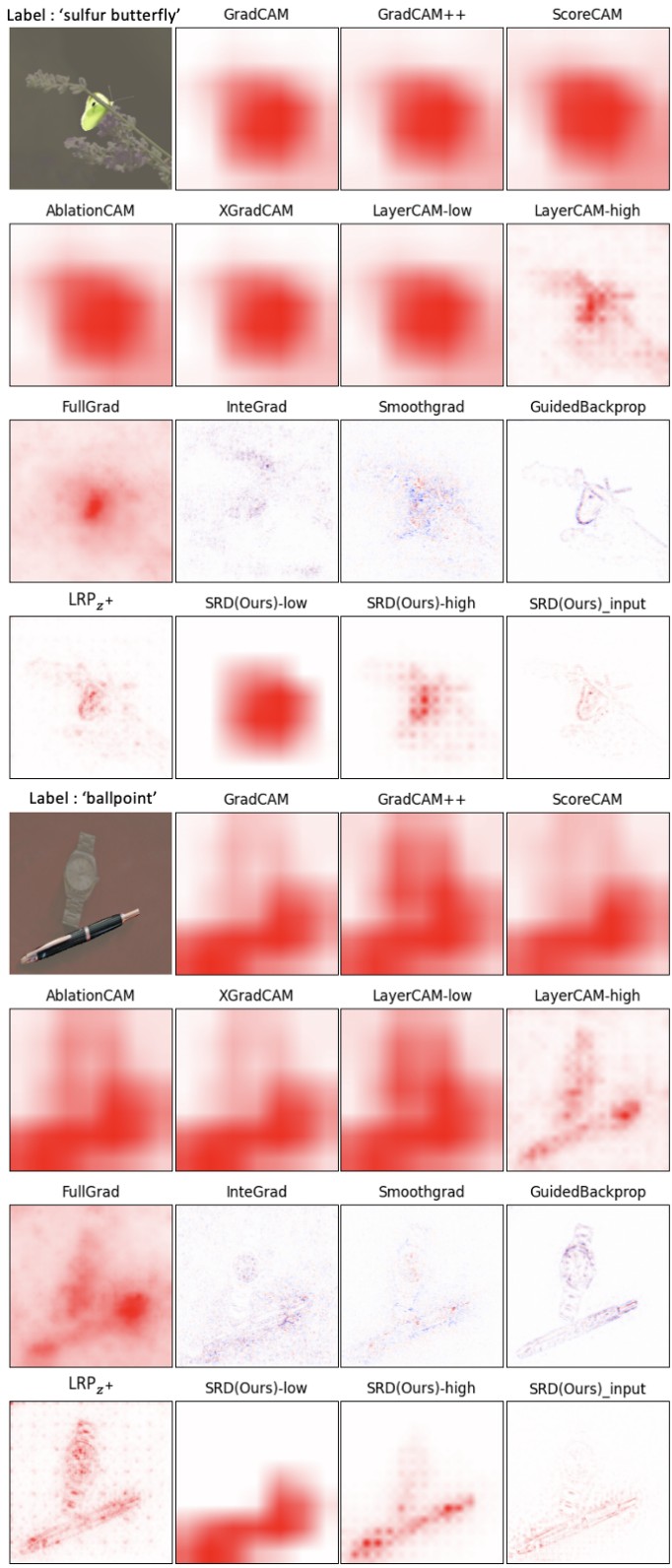

Figure 14: Qualitative comparison on ResNet50. The highlighted region is the segmentation mask.

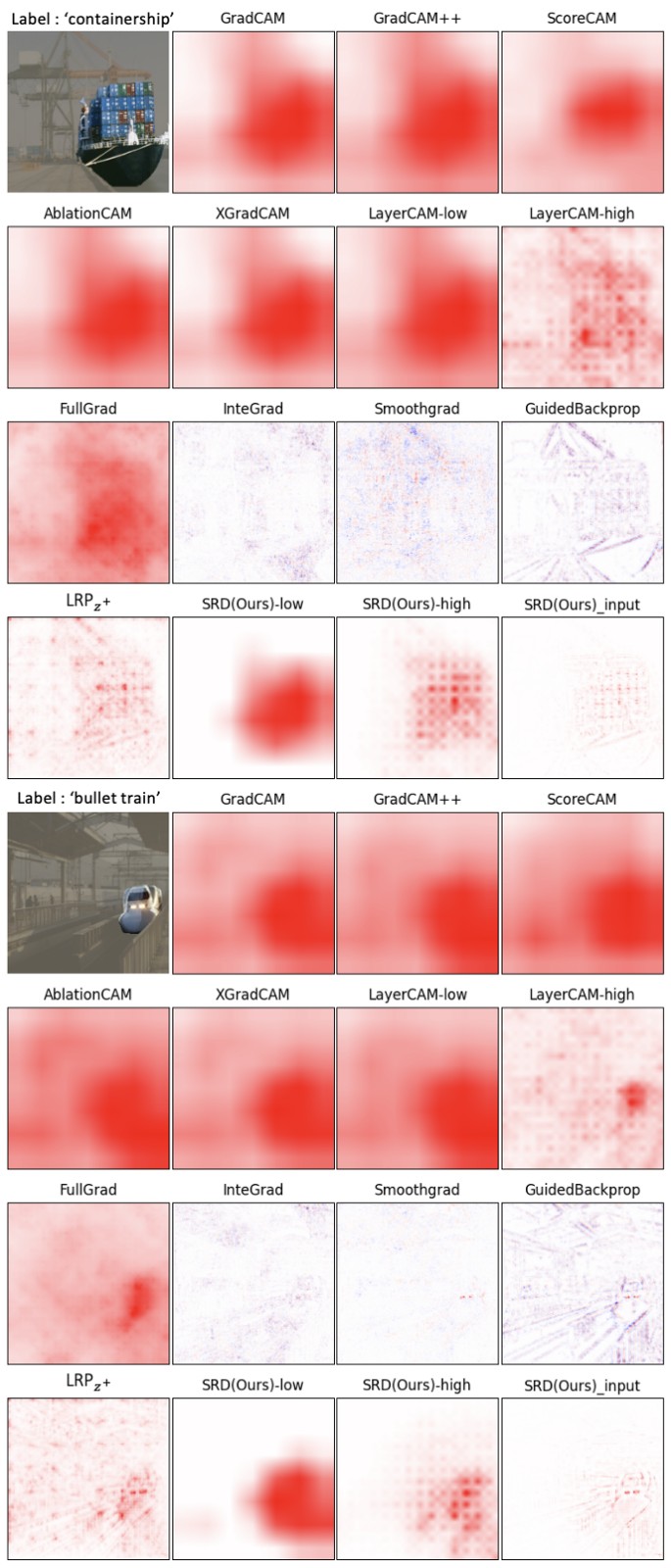

Figure 15: Qualitative comparison on ResNet50. The highlighted region is the segmentation mask.

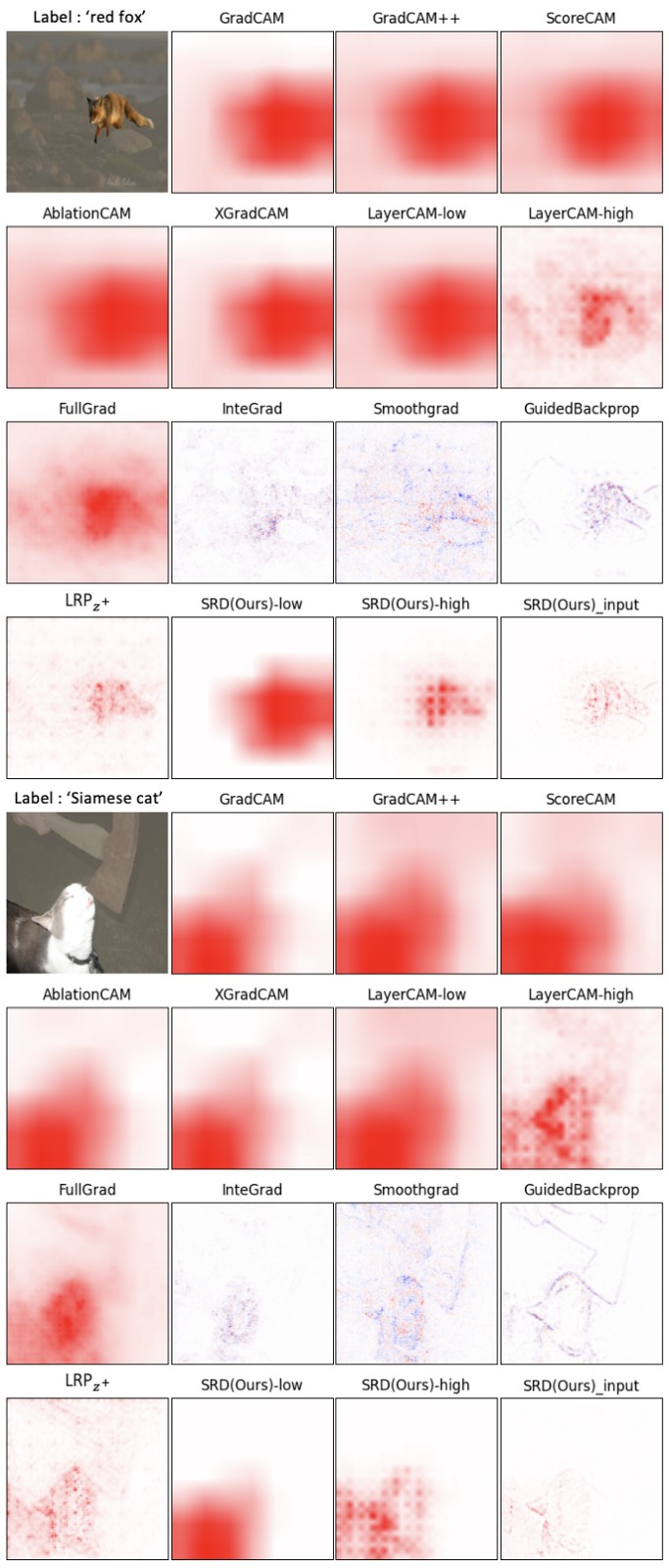

Figure 16: Qualitative comparison on ResNet50. The highlighted region is the segmentation mask.

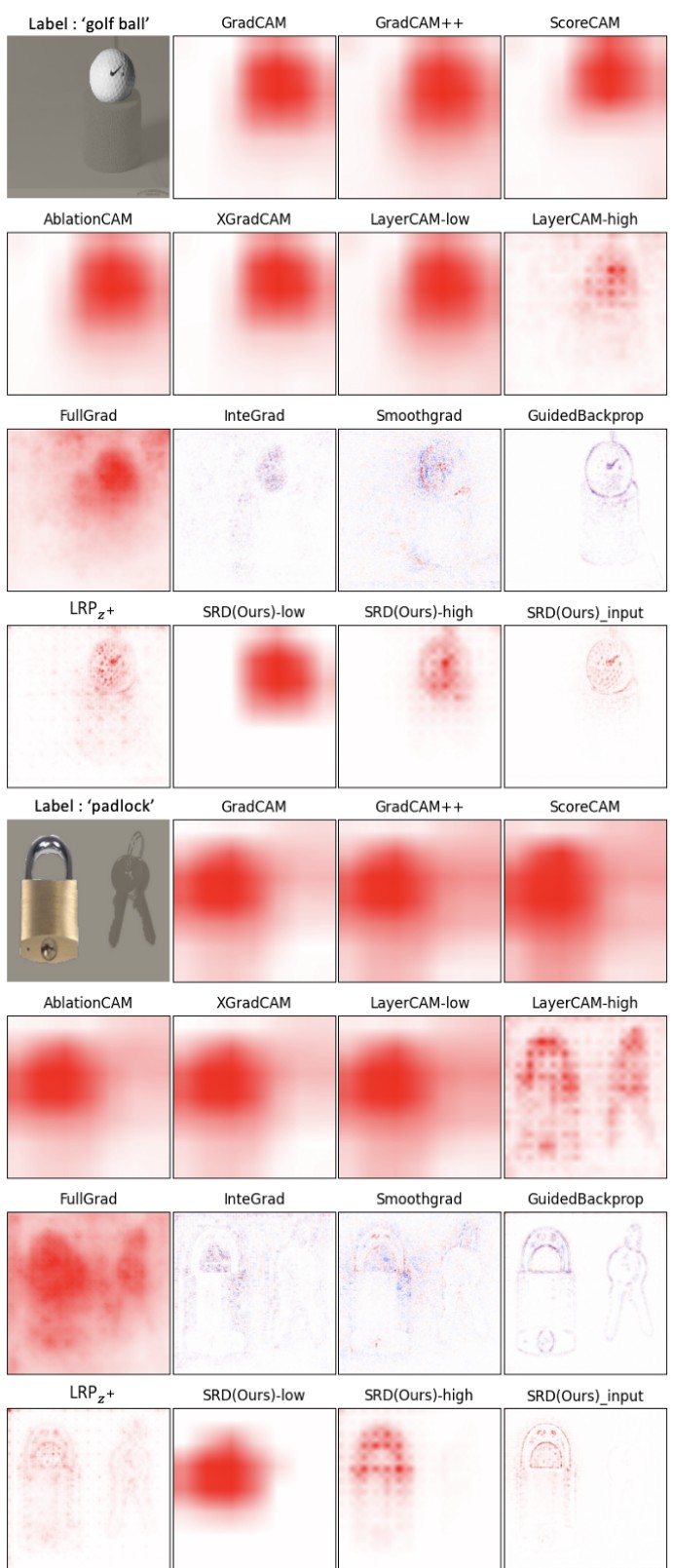

Figure 17: Qualitative comparison on ResNet50. The highlighted region is the segmentation mask.

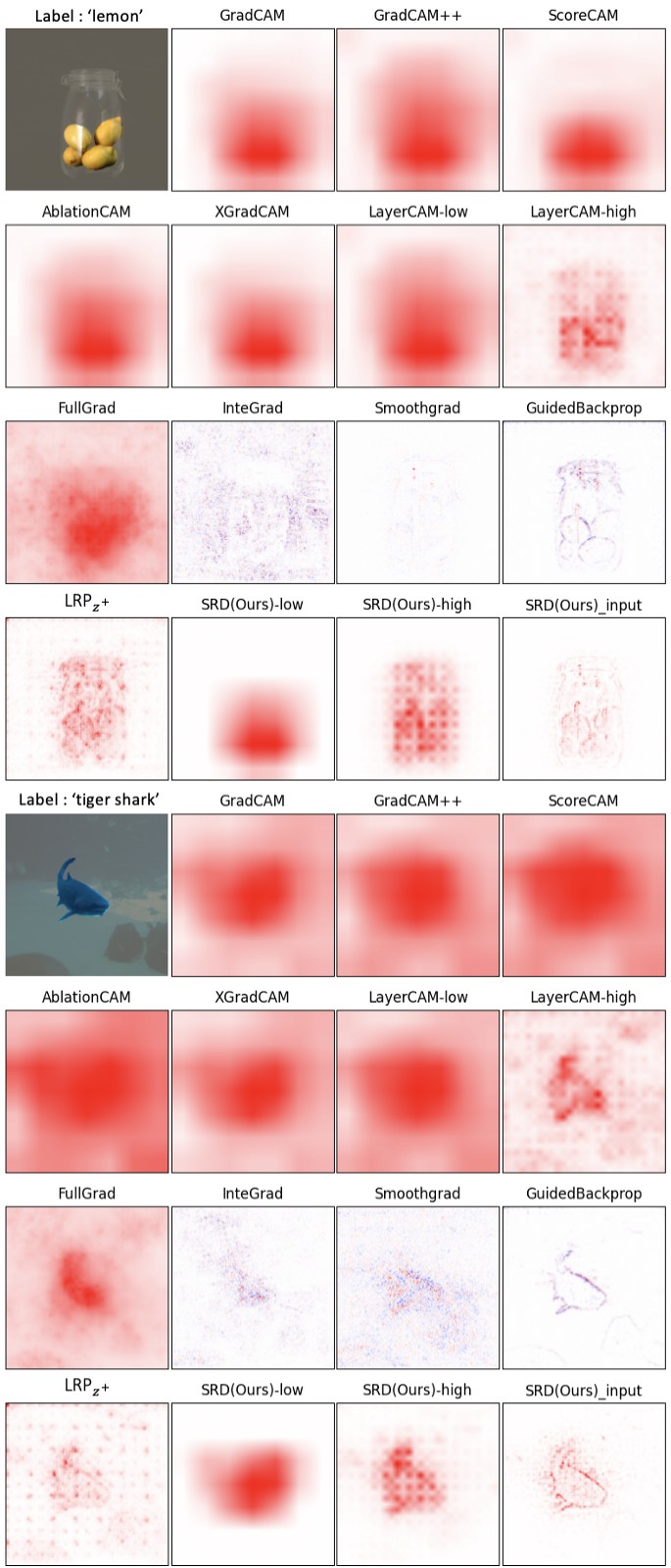

Figure 18: Qualitative comparison on ResNet50. The highlighted region is the segmentation mask.

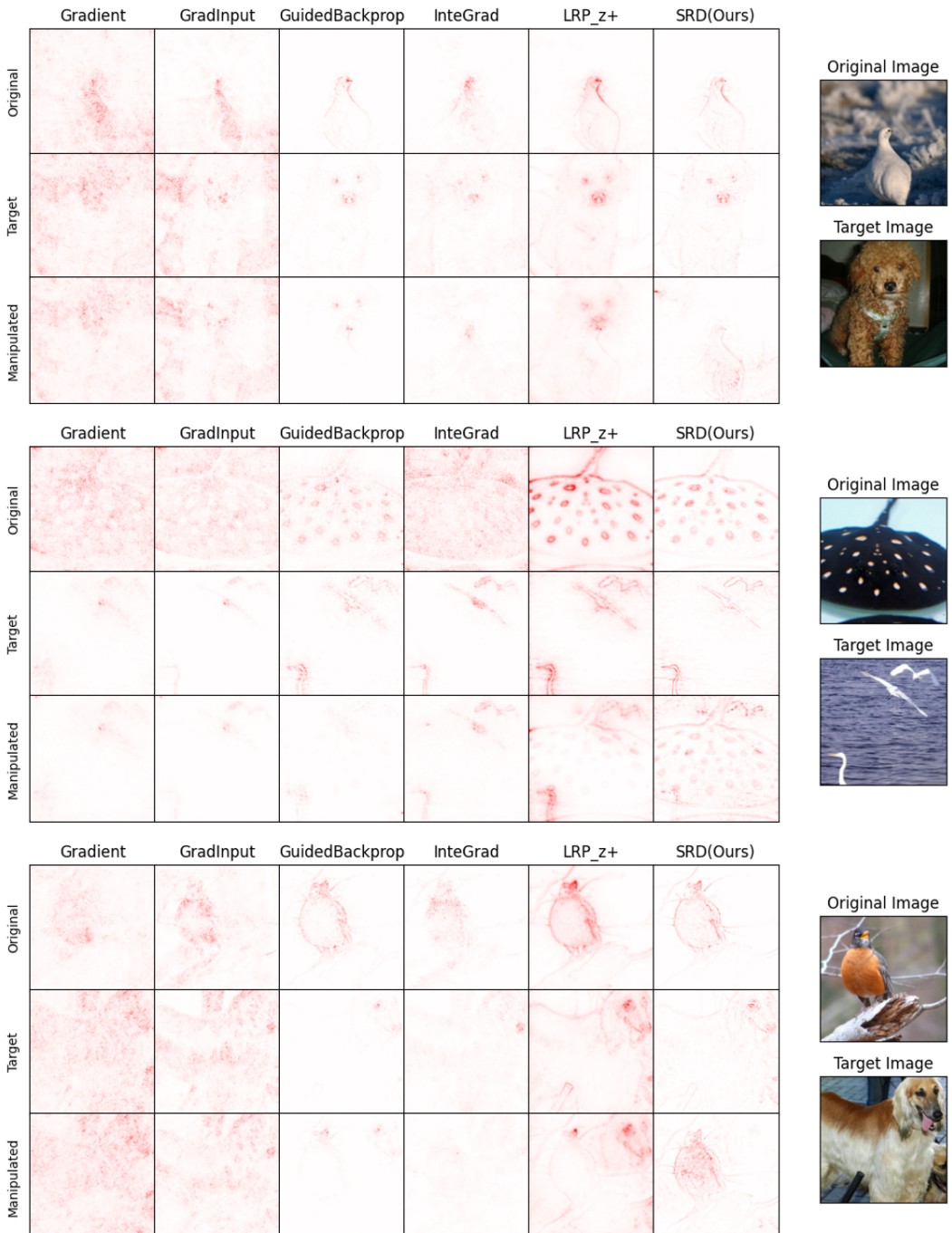

Figure 19: Additional results on explanation manipulation comparison.

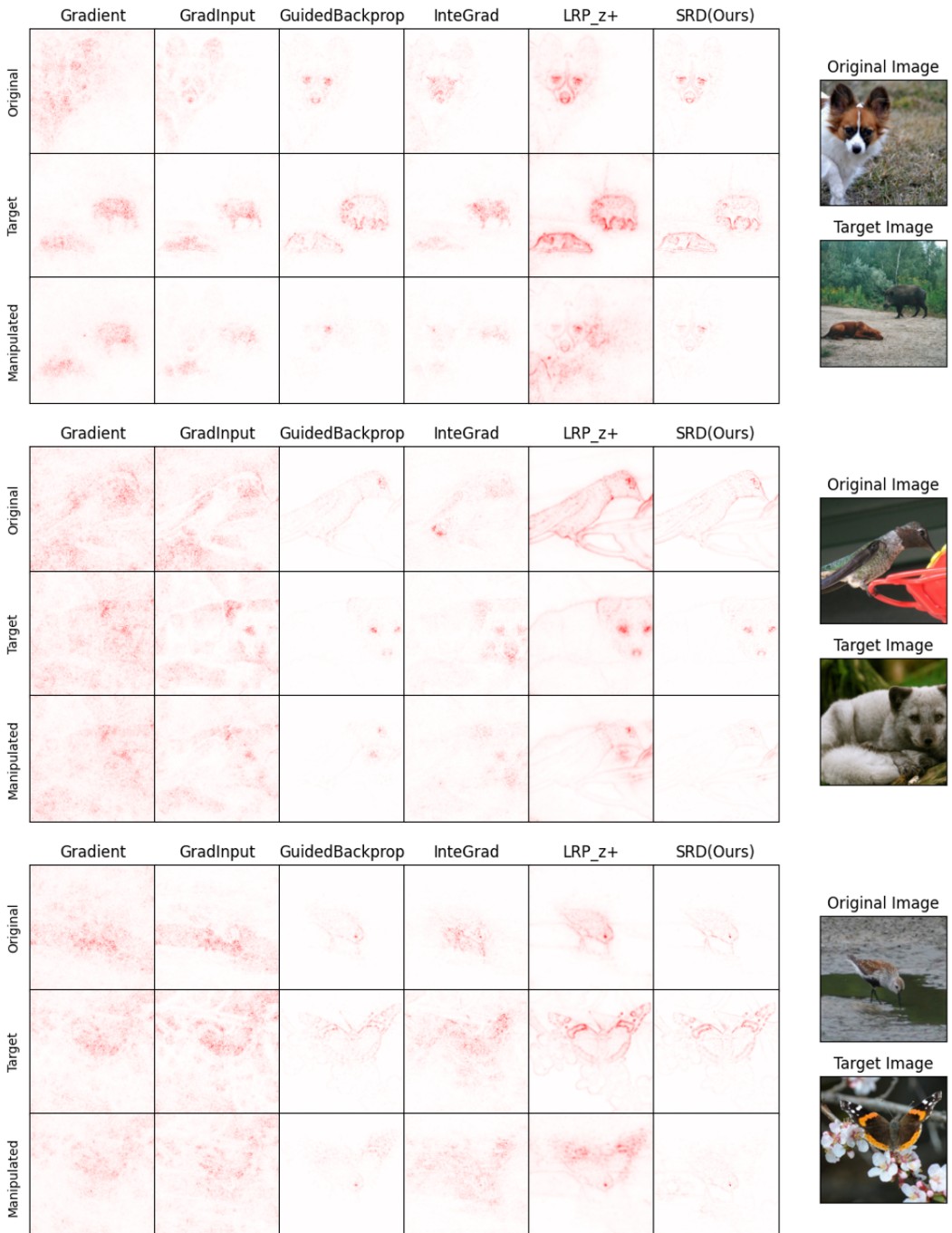

Figure 20: Additional results on explanation manipulation comparison.

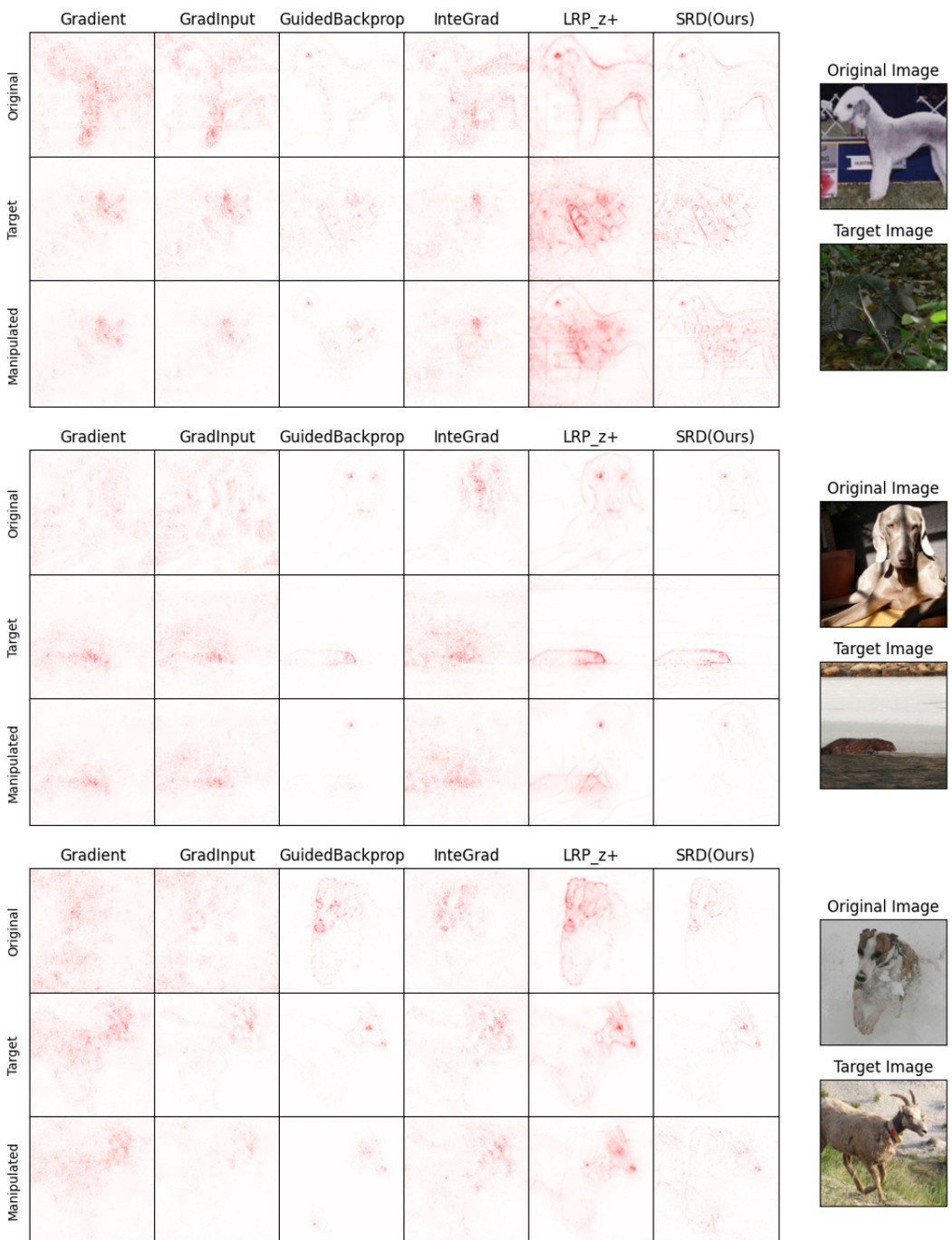

Figure 21: Additional results on explanation manipulation comparison.

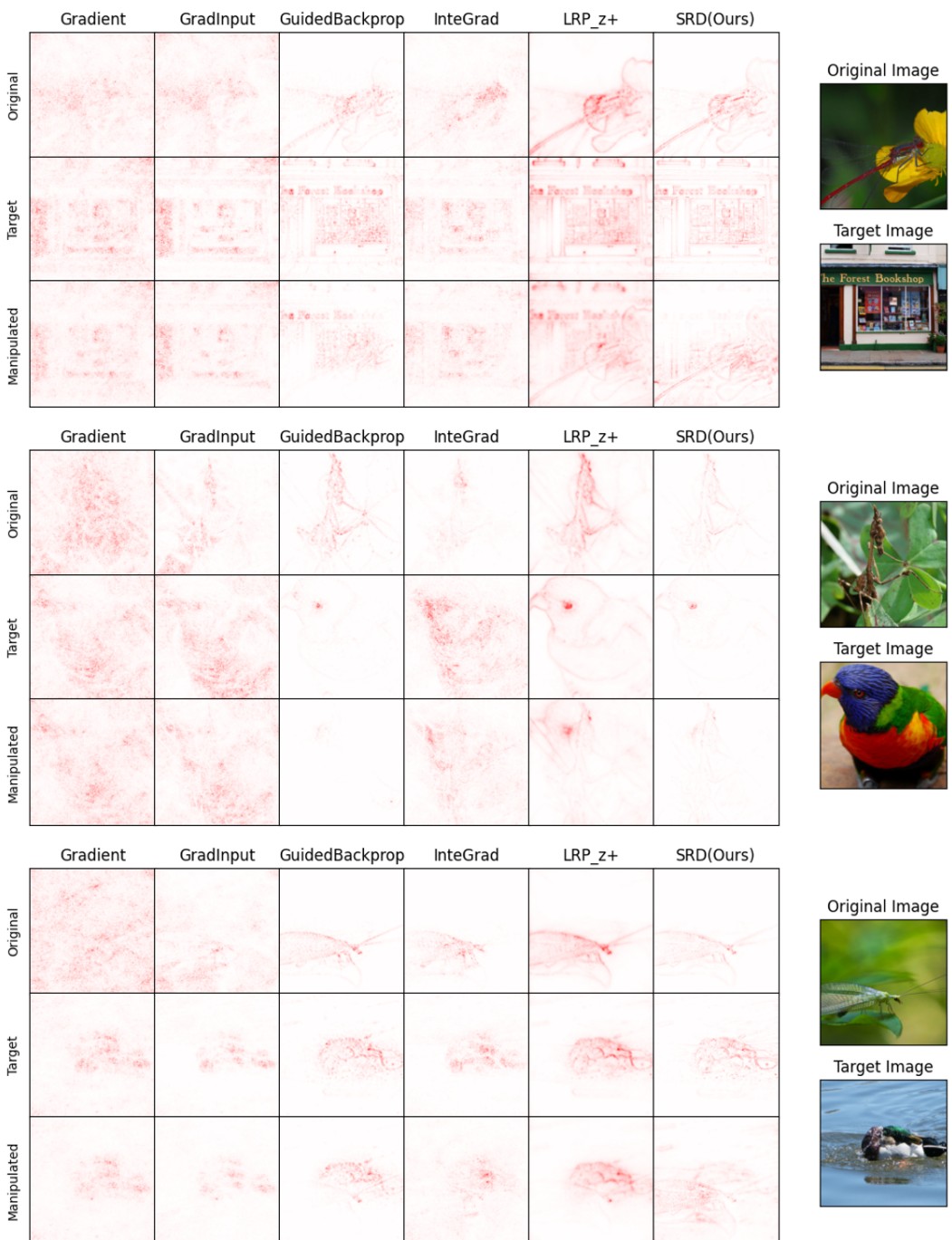

Figure 22: Additional results on explanation manipulation comparison.

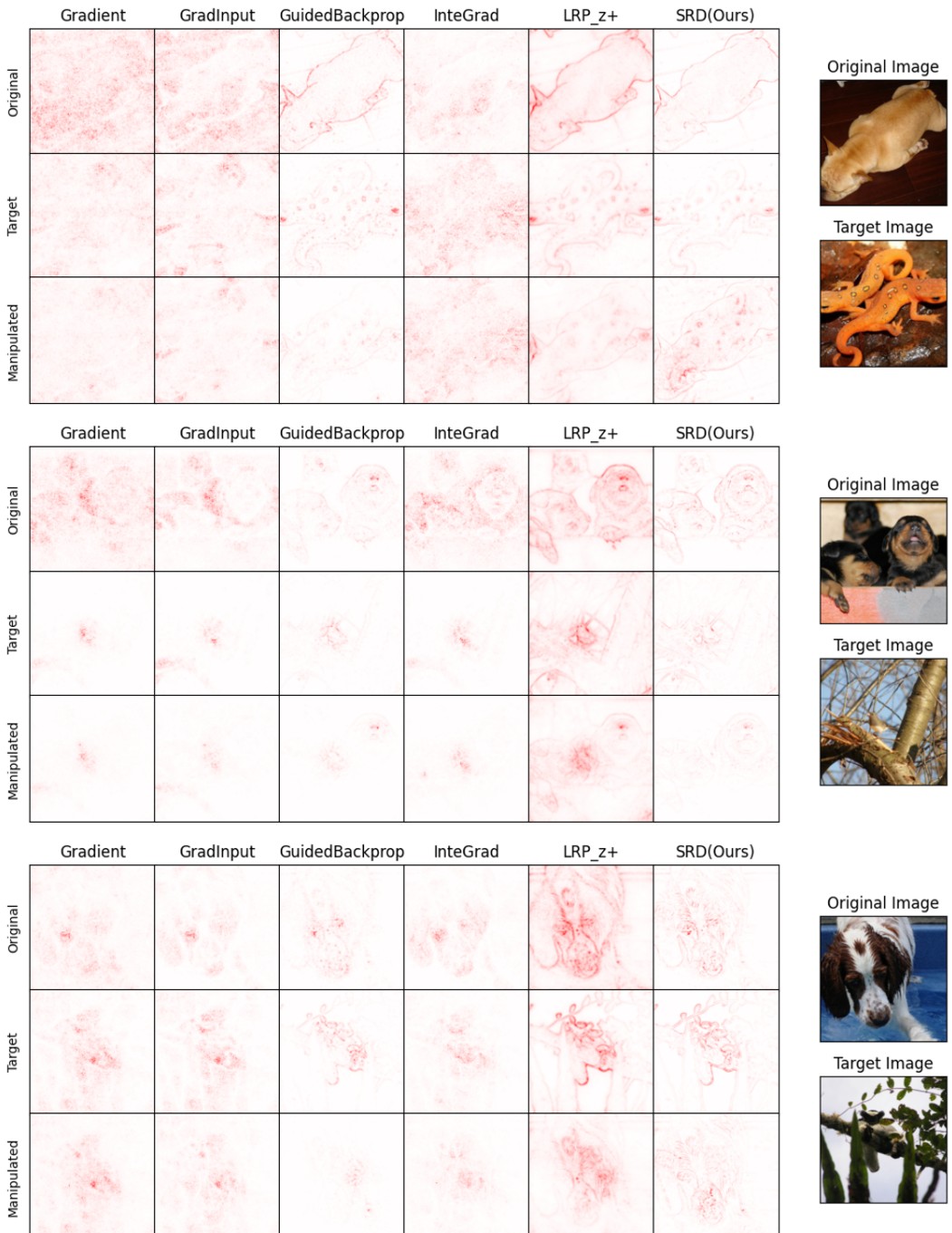

Figure 23: Additional results on explanation manipulation comparison.

