# OpenReview forum: "Respect the model: Fine-grained and Robust Explanation with Sharing Ratio Decomposition"
_ICLR.cc/2024/Conference — ICLR 2024 poster_

### Official Review · Reviewer_HQ8j · 2023-10-27

**Soundness:** 2 fair
**Presentation:** 3 good
**Contribution:** 3 good
**Rating:** 6
**Confidence:** 2

**Summary:**

The authors propose a novel eXplainable AI (XAI) method called SRD (Sharing Ratio Decomposition), which sincerely reflects the model’s inference process, resulting in significantly enhanced robustness in our explanations. Through experimental validation, this approach has yielded an intriguing result.

**Strengths:**

The writing structure is well-organized, the language is fluent, and it exhibits strong readability. The inclusion of rich visualizations and thorough experimental validation adds to its overall quality. In the scheme design, different from the conventional emphasis on the neuronal level, they adopt a vector perspective to consider the intricate nonlinear interactions between filters.

**Weaknesses:**

For the entire paper, the organization of the content is not very reasonable. There is limited coverage in the experimental description, whereas the related work section is extensive. The results in the experiments need more in-depth analysis and discussion.

All evaluations are performed on the ImageNet-S50 dataset, and it would be better to conduct evaluations on multiple datasets to better demonstrate the proposed method's generalization capabilities.

The experimental comparisons and performance improvements lack strong persuasiveness, and the conclusion section of the paper is lengthy and somewhat verbose.

**Questions:**

In the experiments, it is not clearly explained why the input resolutions vary between different methods. Furthermore, it's not clear how using different resolutions as inputs ensures the fairness of the experiments.

The existing methods have deviated from faithfully representing the model, making them susceptible to adversarial attacks. The author did not provide relevant discussions on this issue.

All the comparative methods used in the experiments are from 2014 to 2021, but it's not clear why more recent methods were not included.

The choice to target the Conv5_3 layer in VGG16 and the avgpool layer in ResNet50 is not adequately justified. Whether adjacent layers would have a significant impact is not addressed.

---

> ### Author Response · Authors · 2023-11-14
>
> First of all, we appreciate the careful consideration given to our work. Below, we provide detailed responses to each of your queries. We hope this helps, but please let us know if you need anything else or if you have any other questions.
>
> > ### W1, W3 (Organization)
> In response to your concern about the organization, we would like to shorten our related work and conclusion so that we could allocate sufficient content for our in-depth experiment analysis. Thank you for your constructive suggestion!
>
> > ### W2 (Experiment; Dataset)
> Regarding the experiment dataset, we used the ImageNet-S50 dataset since we need a segmentation mask to evaluate the precise localization. Yet, the versatility of our method is truly remarkable, extending across various datasets, tasks, and even, across all activations **(Extra > CIFAR 100 > Activation_performance_table.png)** due to the utilization of preactivation. Consequently, it is expected to perform well on conventional networks. While implementation complexities led to its application primarily in VGG and ResNet architectures, it has proven effective even in reinforcement learning scenarios in simple conv net. **(Extra > DQN > gif and png files)**
>
> > ### Q1 (Experiment; settings)
> For question 1 (resolution difference in our experiment), the input resolutions are the same, but the resolution of the generated explanations are different. As stated in our paper (experiment), we used different resolutions since most of the CAM methods do not work well at shallow layers. Therefore, to compare with their best performances, we conducted our experiments in various resolutions.
>
> > ### Q2 (Motivation; Adversarial Attacks)
> For question 2 (adversarial attacks in XAI), our intention to that statement was that since most methods utilize the external information including gradients or SHAPLEY values, they fail to explain the adversarial attacks [5]. We would modify our statement and include references to justify our statement.
> [5] Baniecki, Hubert, and Przemyslaw Biecek. "Adversarial Attacks and Defenses in Explainable Artificial Intelligence: A Survey." arXiv preprint arXiv:2306.06123 (2023).
>
> > ### Q3 (Experiment; Latest Methods)
> For question 3 (latest methods), we also aspired to undertake comparative experiments with the most recent methodologies; however, regrettably, access to the corresponding source codes is presently unavailable *(We sent mails to the authors.)*. This circumstance elicits our sincere apologies.
>
> > ### Q4 (Experiment; Settings)
> For question 4 (justification of targeting the layers), it is because the CAM-based methods have the best performance in those layers, using those layers as their baselines *(even in libraries, pytorch-grad-cam(https://github.com/jacobgil/pytorch-grad-cam), torchcam(https://github.com/frgfm/torch-cam))*. It is stated in our paper, but we are sorry for making any confusion by providing too detailed experiment settings. We would revise this experiment setting part and put this into the Appendix.

---

> ### Comment · Reviewer_HQ8j · 2023-12-05
>
> Part of my concerns are addressed. Therefore, I do not change my rating.

---

### Official Review · Reviewer_WQ6F · 2023-10-31

**Soundness:** 3 good
**Presentation:** 3 good
**Contribution:** 3 good
**Rating:** 6
**Confidence:** 3

**Summary:**

The authors introduce a novel interpretability method in this paper. Their approach differs from the past neural perspectives and starts from a vector perspective, leveraging gradient information and intermediate model output propagation computations. They attribute the input images based on the model’s inference process and explain the algorithm’s computation process from both forward and backward equivalent processes.

**Strengths:**

1.  The authors propose a new perspective with finer granularity, attributing based on vectors as units.
2.  The authors conducted experiments from aspects such as Localization, Complexity, Faithfulness, and Robustness based on previously proposed desiderata of interpretability, demonstrating the effectiveness of the algorithm.
3.  Additionally, the authors introduce an interesting observation worth further research, Activation-PatternOnly Prediction (APOP), indicating that retaining only the activation states in the network can still preserve the model’s inference capability to a certain extent.

**Weaknesses:**

1.  The experiments in this paper are all based on the ImageNet-S50 dataset, which only contains 752 samples. To better verify the effectiveness of the algorithm, it would be more persuasive to test on the ImageNet validation set like perturbation[1] and deletion & insertion[2] experiments do.
 2.  From my personal point of view, the APOP phenomenon is a very interesting and worthwhile topic for further research. However, I didn’t quite grasp how the APOP phenomenon helps with attribution in the model’s inference process. That is, this section doesn’t seem to be tightly connected with the paper’s main focus, the interpretability algorithm.
 3.  The experiments in this paper are only based on VGG and ResNet backbones. Perhaps conducting experiments on backbones of different depths and comparing the algorithm’s performance on shallow layers and deep layers would make the paper more convincing.

[1] VU MinhN, NGUYEN TrucD T, PHAN N, et al. Evaluating Explainers via Perturbation[J]. arXiv: Learning,arXiv: Learning, 2019.

[2] PETSIUK V, DAS A, SAENKO K. RISE: Randomized Input Sampling for Explanation of Black-box Models[J]. arXiv: Computer Vision and Pattern Recognition,arXiv: Computer Vision and Pattern Recognition, 2018.

**Questions:**

1. The paper's experiments should include testing on the larger ImageNet validation set for better verification.
2. The APOP, although interesting, doesn't seem connected to the motivation that focus on interpretability.
3. The paper should consider conducting experiments with different backbone architectures to strengthen its findings.

---

> ### Author Response · Authors · 2023-11-14
>
> First of all, we would like to express our gratitude for your thorough evaluation. Below, we provide detailed responses to each of your queries. We hope this helps, but please let us know if you need anything else or if you have any other questions.
>
> > ### Q1 (Experiment; Dataset)
> Regarding the experiment dataset, we used the ImageNet-S50 dataset since we need segmentation masks to precisely evaluate the attribution localization. We believe that this dataset is large enough for evaluating explanation methods, since this dataset contains 752 samples, where [3] used 700 images in the Caltech101 dataset, and [4] used 500 samples in the MNIST/CIFAR-10 dataset and 100 samples in the ImageNet dataset.
> [3] VU MinhN, NGUYEN TrucD T, PHAN N, et al. Evaluating Explainers via Perturbation[J]. arXiv: Learning,arXiv: Learning, 2019.
> [4] Fel, T., Ducoffe, M., Vigouroux, D., Cadene, R., Capelle, M., Nicodeme, C., & Serre, T. (2022). Don't Lie to Me! Robust and Efficient Explainability with Verified Perturbation Analysis. 2023 IEEE/CVF Conference on Computer Vision and Pattern Recognition (CVPR), 16153-16163.
>
> > ### Q2 (Motivation; APOP)
> Also, regarding the connection between the APOP phenomenon and our method, we acknowledge the inadequacy of our previous explanation and will provide a revised version to convey our motivation of using preactivation values. As shown in Table 1, APOP shows that classification accuracies can be maintained with only the on/off activation pattern of the network. This emphasizes the influence of activation patterns, including inactive neurons which contribute to making these patterns. Thank you for your constructive suggestion!
>
> > ### Q3 (Extension of our method)
> Lastly, the versatility of our method is truly remarkable, extending across various datasets, tasks, and even, across all activations **(Extra > CIFAR 100 > Activation_performance_table.png)** due to the utilization of preactivation. Consequently, it is expected to perform well on conventional networks. While implementation complexities led to its application primarily in VGG and ResNet architectures, it has proven effective even in reinforcement learning scenarios in simple conv net. **(Extra > DQN > gif and png files)**

---

### Official Review · Reviewer_z79h · 2023-11-02

**Soundness:** 2 fair
**Presentation:** 2 fair
**Contribution:** 3 good
**Rating:** 6
**Confidence:** 3

**Summary:**

This paper proposes an XAI method called Sharing Ratio Decompostion at vector level. It can reflect the inference process at the vector level instead of neuron level. It also proposes a new observation method called APOP which can highlight the influence of the inactive neurons. The result on ImageNet dataset reach outstanding performance compared to SOTA.

**Strengths:**

1. The quantitative experiment result is good.
2. APOP is a novel observation method to highlight the influence of inactive neurons and easy to follow.
3. The deduction of SRD on forward and backward pass is rigorous.

**Weaknesses:**

1. Compared SOTA is limited and all methods were proposed before 2021. There are many CAM based methods in these two years.Why do not compare to the methods in these two years?
2. In equation 4, it mentions that the summation of modified sharing ratio is not 1. So will it cause some problem? It seems better to normalize ratio to 1 further.
3. The metrics used in experiment is quite different to other XAI methods. Is there any consideration in that?

**Questions:**

Questions can be seen in weakness.

---

> ### Author Response · Authors · 2023-11-14
>
> First of all,  we express our gratitude for acknowledging our motivation and works. Below, we provide detailed responses to each of your queries. We hope this helps, but please let us know if you need anything else or if you have any other questions.
>
> > ### Q1 (Latest methods)
> Regarding the query in comparison with the latest methods, we also aspired to undertake comparative experiments with the most recent methodologies; however, regrettably, access to the corresponding source codes is presently unavailable *(We sent mails to the authors.)*. This circumstance elicits our sincere apologies.
>
> > ### Q2 (Method; Modified Sharing Ratio)
> For the question on the modified sharing ratio, we can normalize it to 1 since the outcomes remain unaffected. Yet, since we employed alternative XAI methods for the MLP layer, we tried to use the outputs without any modification.
>
> > ### Q3 (Experiment; Metrics)
> And we are also aware that Insertion/Deletion is a widely used metric, falling under the category of Faithfulness. However, as in [2], with Insertion/Deletion, it remains ambiguous whether the drop in model performance results from a shift in data distribution or from the deletion/insertion of the informative features. For example, even though InputXGrad does not seem faithful, it exhibits excellent deletion performance **(ResNet50: 0.0842, VGG16: 0.0459)**. This may be attributed to the fact that logit values diminish rapidly as input quickly falls into out-of-distribution. Likewise, in the case of Insertion, explanation methods with low-resolution demonstrate superior performance, since it is advantageous to just fill the area as fast as possible, independently of the model’s explanation. Conversely, the Fidelity metric (which we used), is relatively more versatile than deletion *(addressing OOD issues)* and insertion *(area filling)* since it involves removing random pixels and observing the change in logits. Hence, we opted for Fidelity in our evaluation, instead of Insertion and Deletion.
> [2] Hooker et al. A Benchmark for Interpretability Methods in Deep Neural Networks, 2019.

---

### Official Review · Reviewer_aoME · 2023-11-10

**Soundness:** 3 good
**Presentation:** 2 fair
**Contribution:** 3 good
**Rating:** 6
**Confidence:** 3

**Summary:**

In this paper, the authors propose an explanation method for deep models called Sharing Ratio Decomposition (SRD) that pursues enhanced precision and robustness of its explanation. In extensive simulations, the method shows a strong and robust explanation performance across various kinds of metrics beyond the prior XAI methods. The main contribution of this work is to widen the view of XAI that should be pursued for achieving a reliable and robust explanation of deep model inference.

**Strengths:**

**Strength 1:** Explainable AI (XAI) is one of the most important research topics that directly tackle the explanation issue of the current deep learning models. Beyond the better explainability, the authors raise the multiple objectives that XAI should pursue, i.e., Localization, Complexity, Faithfulness, and Robustness, and argue that the proposed XAI, called Sharing Ratio Decomposition (SRD), is superior to the prior XAIs in the 'desiderata'. I am strongly convinced that the future XAI should consider an efficient, effective, precise, and robust explanation by pursuing the desired factors. The comprehensive experimental results in Table 2 clearly show the strength of the proposed XAI method, which is SRD.

**Strength 2:** A lot of qualitative results in the main paper and the supplementary truly help readers visually understand the benefits of SRD. These results clearly contrast SRD to the prior methods. I believe that efforts to show these qualitative results are very important to visually identify the explainability that often relies on the human-concept level evaluation of how well XAI explains the model's inference.

**Strength 3:** A bunch of prior works are clearly compared to the proposed XAI method via extensive simulation results (as shown in Table 2). I expect that the evaluation effort will promote extensive comparison with diversified viewpoints, i.e., Poi., Att., Spa., Fid., and Sta., in assessing future XAIs.

**Weaknesses:**

**Weakness 1:** The '3. Method' part needs to be improved for better presentation. For readers who are a beginner in the XAI research field, I feel that a compact and clear description such as a pseudocode-style presentation or stepwise procedure explanation (e.g., 1. Feedforward an input, 2) Calculate PFV, 3) Decompose PFV, 4) Compute sharing ration through backward propagation, 5) Relevance is computed), would definitely help readers to understand the way that the method works. In this version of the article, the description is a quite verbal explanation.

**Weakness 2:** The key reasons behind the gains of SRD are not clearly analyzed. In this paper, the authors have separately pointed out the uniqueness of their method in multiple parts of the paper. However, I cannot fully understand the essential factors that make SRD be superior to other prior XAI methods. Is it due to the vector perspective, consideration of inactive neurons, decomposition of PFV, or any combinatorial effect of multiple factors?

**Questions:**

**Q1:** In the 'Weakness 1' issue, would you provide a stepwise description of the methodology?

**Q2:** In the 'Weakness 2' issue, would you clarify the main reason of the gains of SRD?

---

> ### Author Response · Authors · 2023-11-14
>
> First of all, thank you for your insightful feedback. We appreciate your agreement to the importance of XAI with your acknowledgement of our work. Below, we provide detailed responses to each of your queries. We hope this helps, but please let us know if you need anything else or if you have any other questions.
>
> > ### Q1 (Method; Beginner-friendly Description)
> Regarding the suggestion for a beginner-friendly explanation of the method part, thank you for your constructive suggestion. We provide forward pass with backward pass, because we thought that our forward pass is intuitive for beginners to understand well and our backward pass is familiar for others who have experience in this field. Yet, we would revise our paper by providing the pseudocode or stepwise procedure explanation for better understanding. For better understanding of our method, please refer to the stepwise procedure description as below. We hope that our stepwise description and the formulas in our paper help you understand our method easier.
> >	> #### **[forward pass]**
> >	> 1. Feedforward an input.
> 2. With the sharing ratio, construct ERF from the first layer. (ERF for an input is the pixel itself as in Eq.(2)) Now, every PFV on every layer has its own assigned ERF.
> 3. Generate the final saliency map with the weighted sum of ERFs assigned for final layer PFVs. (Eq. (3))
> >	> #### **[backward pass]**
> >	> 1. Generate PFVs as in Eq.(5).
> 2. Compute sharing ratio as in Eq.(6).
> 3. Decompose the relevance to the output as in Eq.(7).
> 4. Generate the final saliency map by summing the relevance shares in its projective field. (Eq.(8))
>
> > ### Q2 (Motivation; Ablation)
> Also, to clarify the gains of our method, we would provide an ablation study for each factor in the supplementary material **(Extra > Ablation > image files)**. The scalar-based approach with our method can be regarded as LRP-0 [1]. And next to it, we present the generated explanation when calculating the relevance with post-activation values. As you can see, compared to ours (SRD), the generated explanations with scalar are very noisy, while those with post-activation value are too sparse. With our observation of APOP, we have proven that we should consider every information including active and inactive neurons. This is the reason that we used vectors as our analysis unit and pre-activation values to propagate our relevance.
> [1] Bach et al. On Pixel-Wise Explanations for Non-Linear Classifier Decisions by Layer-Wise Relevance Propagation, 2015.

---

> > ### Comment · Reviewer_aoME · 2023-11-23
> > **Thanks for the response**
> >
> > Dear Authors,
> >
> > I appreciate your kind response and additional qualitative results.
> > I hope you to add your procedure description of SRD in the main manuscript, if accepted.
> > Also, please add the additional qualitative results to the paper, at least supplementary.
> > Moreover, I want to value the extension of SRD that you highlighted during the rebuttal, i.e., applicable to different activations and RL tasks.
> >
> > I raised my rating to 6, above the acceptance level.
> > Again, thanks for the efforts to address the raised issues.

---

> > > ### Author Response · Authors · 2023-11-23
> > > **Thank you for acknowledging our contribution.**
> > >
> > > Thank you for acknowledging our contribution and thank you again for your detailed review and helpful suggestions. We will make sure to include the additional results in our final version.

---

### Author Response · Authors · 2023-11-14
**For all reviewers: supplementary materials**

We upload our supplementary materials for better understanding of our methods.
> ## Ablation study
To clarify the gains of our method, we would provide an ablation study for each factor in the supplementary material **(Extra > Ablation > image files)**. The scalar-based approach with our method can be regarded as LRP-0 [1]. Next to it, we present the generated explanation when calculating the relevance with post-activation values. As you can see, compared to ours (SRD), the generated explanations with scalar are very noisy, while those with post-activation values are too sparse. With our observation of APOP, we have proven that we should consider every information including active and inactive neurons. This is the reason that we used vectors as our analysis unit and pre-activation values to propagate our relevance.
 [1] Bach et al. On Pixel-Wise Explanations for Non-Linear Classifier Decisions by Layer-Wise Relevance Propagation, 2015.

> ## Extension of our method
The versatility of our method is truly remarkable, extending across various datasets, tasks, and even, across all activations **(Extra > CIFAR 100 > Activation_performance_table.png)** due to the utilization of preactivation. Consequently, it is expected to perform well on conventional networks. While implementation complexities led to its application primarily in VGG and ResNet architectures, it has proven effective even in reinforcement learning scenarios in simple conv net. **(Extra > DQN > gif and png files)**

---

### Meta-Review · Area_Chair_AXQV · 2023-12-12

**Metareview:**

This paper introduces a novel explanation method for deep models called Sharing Ratio Decomposition (SRD), which aims to improve the precision and robustness of its explanations. The results of extensive simulations demonstrate that SRD outperforms existing XAI methods across various metrics, showing strong and robust explanation performance. While the paper addresses an important research topic and provides new insights, several reviewers have pointed out that the evaluation section could be further improved. Despite this limitation, the proposed method shows promise as a valuable tool for understanding and interpreting the behavior of deep learning models.

**Justification For Why Not Higher Score:**

N/A

**Justification For Why Not Lower Score:**

The proposed method shows promise as a valuable tool for understanding and interpreting the behavior of deep learning models.

---

### Decision · Program_Chairs · 2024-01-16

Accept (poster)